# You Only Train Once: Differentiable Subset Selection for Omics Data

**Daphné Chopard**                                                      *daphne.chopard@inf.ethz.ch*
*Department of Computer Science, ETH Zurich*
*Department of Intensive Care and Neonatology, University Children's Hospital Zurich*
**Jorge da Silva Gonçalves**                                      *jorge.dasilvagoncalves@inf.ethz.ch*
*Department of Computer Science, ETH Zurich*
**Irene Cannistraci**                                                  *irene.cannistraci@inf.ethz.ch*
*Department of Computer Science, ETH Zurich*
**Thomas M. Sutter**[*]                                                  *thomas.sutter@inf.ethz.ch*
*Department of Computer Science, ETH Zurich*
**Julia E. Vogt**[*]                                                        *julia.vogt@inf.ethz.ch*
*Department of Computer Science, ETH Zurich*

**Reviewed on OpenReview:** *https://openreview.net/forum?id=xQiXlADW5v*

## Abstract

Selecting compact and informative gene subsets from single-cell transcriptomic data is essential for biomarker discovery, improving interpretability, and cost-effective profiling. However, most existing feature selection approaches either operate as multi-stage pipelines or rely on post hoc feature attribution, making selection and prediction weakly coupled. However, most existing feature selection approaches either operate as multi-stage pipelines or rely on post hoc feature attribution, making selection and prediction weakly coupled. In this work, we present YOTO (you only train once), an end-to-end framework that jointly identifies discrete gene subsets and performs prediction within a single differentiable architecture. In our model, the prediction task directly guides which genes are selected, while the learned subsets, in turn, shape the predictive representation. This closed feedback loop enables the model to iteratively refine both what it selects and how it predicts during training. Unlike existing approaches, YOTO enforces sparsity so that only the selected genes contribute to inference, eliminating the need to train additional downstream classifiers. Through a multi-task learning design, the model learns shared representations across related objectives, allowing different tasks to inform one another, and discovering gene subsets that generalize across tasks without additional training steps. We evaluate YOTO on two representative single-cell RNA-seq datasets, showing that it consistently outperforms state-of-the-art baselines. These results demonstrate that sparse, end-to-end, multi-task gene subset selection improves predictive performance and yields compact and meaningful gene subsets, advancing biomarker discovery and single-cell analysis.[1]

## 1 Introduction

Identifying informative subsets of genes from high-dimensional single-cell transcriptomic data is a fundamental problem in biomedicine (Hwang et al., 2018; Luecken & Theis, 2019). Such gene sets not only reduce measurement costs and technical noise but can also provide mechanistic insight into disease processes and facilitate interpretable predictive models (Heumos et al., 2023). Yet this task remains challenging: single-cell data are high-dimensional, sparse, and noisy and labels are often incomplete or only available for subsets of samples (Luecken & Theis, 2019; Su et al., 2022; He et al., 2022).

---

[*]Shared senior authorship
[1]Code is publicly available: https://github.com/chopardda/yoto.

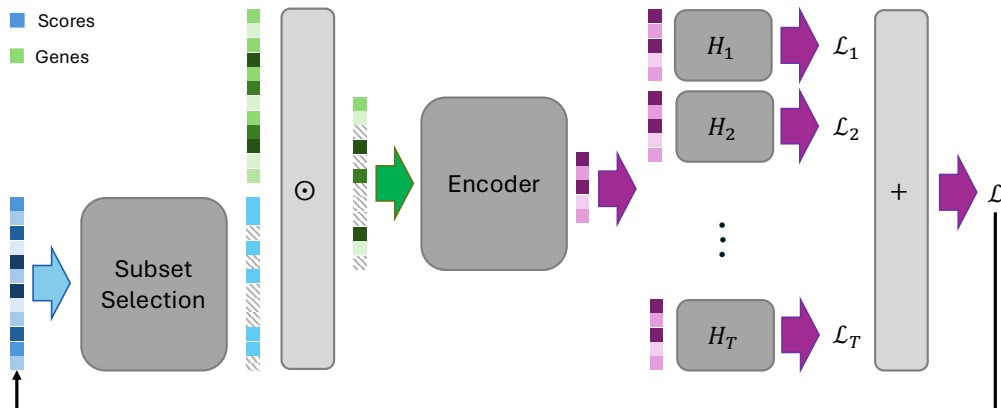

Figure 1: **Overview of YOTO.** Our method, YOTO, consists of three main building blocks: (1) a subset selection block (section 3.1), (2) a shared encoder for all tasks and (3) a multi-task learning block (section 3.2). The subset selection block applies a binary mask that selects the $k$ highest-scoring genes for a given set of tasks, based on a set of learnable scores that are optimized end-to-end using the task loss. The shared encoder encodes the $k$ gene values into a latent representation, which is fed to the task-specific heads.

Traditional approaches for identifying informative genes, including spatial localization (Satija et al., 2015; Stuart et al., 2019), statistical filtering (Ding & Peng, 2003; Peng et al., 2005), and regularized regression (Yamada et al., 2014; Climente-González et al., 2019), have been widely used but are typically decoupled from downstream predictive tasks. As a result, the selected genes may not be optimal for the models they ultimately support. Recent deep learning methods have attempted to address this limitation by integrating feature selection into neural network architectures (Covert et al., 2023), but most rely on post-hoc selection mechanisms that only enforce a sparse selection mask after training and, hence, do not fully constrain the model to a discrete subset of genes during training. Consequently, selection and prediction remain only loosely coupled, and the resulting gene panels can be unstable or require retraining of downstream classifiers for evaluation.

Here, we introduce YOTO (You Only Train Once), an end-to-end gradient-based framework that learns to select subsets and uses the loss signal from prediction tasks to learn these subsets (see Figure 1). YOTO enforces a sparse selection mask during training while being fully differentiable. This establishes a closed feedback loop in which the prediction task guides which genes are selected, while the selected genes refine the model's predictions. By enforcing sparsity, only the selected genes contribute to inference, eliminating the need to retrain separate classifiers and ensuring that evaluation directly reflects the performance of the selected subset. Beyond unifying selection and prediction, our framework incorporates multi-task learning, enabling multiple predictive objectives (e.g., cell type and disease state) to be optimized simultaneously. This design allows YOTO to learn shared gene subsets that generalize across tasks without the need to retrain from scratch for each task. This also means that through a single training instance of YOTO, one can evaluate the performance of the model on all tasks without the need for any additional training. The result is a flexible and data-efficient system that captures both task-specific and shared biological signals.

We evaluate our approach on two distinct single-cell RNA-seq datasets spanning species, tissues, and experimental protocols: one from the mouse primary visual cortex (VISp, Ramsköld et al., 2012) and another from peripheral blood mononuclear cells (PBMCs) of COVID-19 patients (Yao et al., 2021). Across both datasets, we benchmark against widely used feature selection approaches under consistent evaluation settings and varying sizes of gene subsets ranging from 16 to 256 out of thousands of measured genes.

Our results show that the proposed model achieves consistently competitive and often higher F1-scores, AUPRC and accuracy than classical and gradient-based baselines. These findings demonstrate that end-to-end feature selection yields a subset of genes that retain strong predictive power across biological contexts, highlighting its potential for biomarker discovery and translational single-cell applications.

## 2 Related Work

Table 1: **Summary of Methodological Differences Across Gene Selection Approaches.** A comparison of methodological properties between existing gene-selection approaches and YOTO, including supervision, task type, classifier integration, sparsity of the selection mechanism, and end-to-end trainability.

| Method | Supervised | Task Type | Built-in Classifier | Sparse Selection | End-to-end |
|---|---|---|---|---|---|
| Seurat | ✗ | - | ✗ | ✓ | ✗ |
| Seurat v3 | ✗ | - | ✗ | ✓ | ✗ |
| mRMR-f | ✓ | single-task | ✗ | ✓ | ✗ |
| mRMR-rf | ✓ | single-task | ✗ | ✓ | ✗ |
| Block HSIC Lasso | ✓ | single-task | ✗ | ✓ | ✗ |
| PERSIST | ✓ | single-task | ✓ | ✗ | (✓) |
| **YOTO** | ✓ | multi-task | ✓ | ✓ | ✓ |

Traditional feature selection approaches for omics data rely primarily on statistical testing or information-theoretic measures. Classical methods such as minimum redundancy-maximum relevance (mRMR, Ding & Peng, 2003; Peng et al., 2005) and its convex, kernel-based alternative HSIC Lasso (Yamada et al., 2014) select features based on their dependency with target variables while minimizing redundancy. In contrast to mRMR, HSIC Lasso employs the Hilbert-Schmidt Independence Criterion (HSIC) to measure statistical dependence between variables and applies an $l_1$ regularization term to select a sparse subset of features. Although effective for bulk or low-dimensional data, these approaches often struggle with scalability and nonlinear dependencies inherent in single-cell or spatial transcriptomics datasets. More recent variants such as block HSIC Lasso (Climente-González et al., 2019) improve computational efficiency, making kernel-based feature selection more applicable to modern high-throughput settings.

In the context of single-cell RNA sequencing (scRNA-seq), feature selection is typically framed as marker gene selection, aiming to identify genes that distinguish specific cell types. Widely used toolkits such as Seurat (Satija et al., 2015) and its successor Seurat v3 (Stuart et al., 2019) perform this task through differential expression analysis, comparing expression between one cluster and all others using statistical tests such as the Wilcoxon rank-sum test or Student's t-test. Other differential expression frameworks (including DESeq2 (Love et al., 2014), Limma-voom (Law et al., 2014), and scDD (Korthauer et al., 2015)) model expression differences parametrically or distributionally but often assume homogeneous variance structures that limit their applicability to single-cell data. However, recent large-scale benchmarks have revealed substantial variability in their performance. Pullin & McCarthy (2024) compared 59 marker gene selection methods across 14 scRNA-seq datasets and found that simple statistical tests such as Wilcoxon and t-test remain among the most effective and computationally efficient approaches, often outperforming more complex machine learning methods.

Beyond statistical testing, machine learning approaches such as Random Forest (Breiman, 2001; Rogers & Gunn, 2005) and RelieF (Urbanowicz et al., 2018) identify informative genes through feature importance estimation or neighborhood-based relevance, offering improved robustness at the expense of interpretability. More recently, deep learning methods have been explored for nonlinear feature selection and include gradient-based approaches (e.g., DeepLIFT (Li et al., 2021)) and perturbation-based techniques (e.g., LIME (Ribeiro et al., 2016)). However, these models are not end-to-end: they provide post hoc feature attributions after model training, rather than learning discrete feature subsets as part of the optimization process. Consequently, selection and prediction are decoupled, and the importance scores are continuous, requiring arbitrary thresholds to define gene panels. In practice, most studies retrain a separate classifier on the selected genes to assess their utility, adding computational overhead and preventing integrated optimization. Furthermore, their reliance on gradient sensitivity or perturbations often leads to instability and scalability challenges in large single-cell datasets.

The most relevant prior work to ours is PERSIST (Covert et al., 2023), a gradient–based feature selection framework designed for spatial transcriptomics. PERSIST identifies compact gene panels that optimally reconstruct genome-wide expression profiles from scRNA-seq data, using a differentiable feature selection

layer and a hurdle loss to account for dropout noise. Although primarily applied to spatial transcriptomics, its formulation is broadly applicable across omics domains, including proteomics. However, PERSIST is not strictly end-to-end: gene selection is regulated by a temperature-controlled mask during training. Hence, at the end of training all genes contribute to the task prediction as the involved mask is not sparse. As a result, the selected panel is not truly discrete, and to evaluate its utility, a separate classifier must be retrained on the chosen genes. This two-step process prevents direct assessment of feature utility within the selection model itself, reducing both integration and computational efficiency.

Our work addresses these limitations (see Table 1) by introducing an end-to-end gene selection framework that learns to select a subset of relevant genes by learning to solve a given set of tasks. Unlike prior approaches, our model supports discrete selection, allowing simultaneous feature selection and task prediction. Moreover, it can jointly optimize multiple predictive objectives, enabling it to leverage heterogeneous (and potentially incomplete) supervision signals. This multi-task design allows the model to identify compact, biologically meaningful subsets of genes that generalize across conditions and annotation levels.

## 3 Methods

Our model, YOTO, consists of three main components: a gene selection module, a shared encoder, and task-specific heads. We assume a dataset $\mathcal{X} = \{X^{(i)}\}_{i=1}^N$, where each $X^{(i)}$ represents the gene expression profile (i.e., input features) of cell $i$, and $N$ is the total number of cells. $X_j$ is a vector of gene expression levels of gene $j$ across all cells ($1 \leq j \leq d$, with $d$ being the number of measured genes). We base gene selection on a differentiable ranking procedure that learns a score for each gene and selects the *top* k genes based on these learned scores. The dataset $\mathcal{X}$ is associated with a set of task labels $\mathcal{Y} = \{Y^{(i)}\}_{i=1}^N$, where each $Y^{(i)} = \left[y_1^{(i)}, \ldots, y_T^{(i)}\right]^T$ contains the $T$ supervision signals for cell $i$. Hence, in general, we assume a multitask learning setting for the considered datasets.

The shared encoder maps the subset of selected genes to a latent representation that is fed to the task-specific sub-networks. The task-specific networks, one per task, perform predictions for classification or regression tasks depending on the available labels. The fully differentiable pipeline, including the subset selection, allows end-to-end optimization of both subset selection and multi-task prediction. See Figure 1 for an overview of our method.

### 3.1 Subset Selection

We employ a differentiable feature selection mechanism, inspired by the Differentiable Random Partition Model (DRPM, Sutter et al., 2023a). DRPM leverages the Gumbel-Softmax trick (Jang et al., 2016; Maddison et al., 2016) and the hypergeometric distribution (Sutter et al., 2023b) to define a differentiable random partition model. We use a vector $\boldsymbol{s}$ of learnable scores, where every element $s_j$ is associated with a specific gene $X_j$. We select relevant genes by ranking all genes according to their learnable scores $s_j$. We assume that the size of the subset $k$ is given. Based on the ranking, we select the top-k genes to form our subset of genes. Each score $s_i$ is a learnable parameter that represents the importance of each gene. Hence, the subset corresponds to the $k$ scores with the highest scores $s_i$.

Unlike previous works on gene selection approaches (e.g., Covert et al., 2023), we explicitly learn the ranking between features, rather than treating them as independent. To enable trainable feature selection while preserving differentiability, we use a differentiable permutation matrix $\pi(\boldsymbol{s})$ (Grover et al., 2019), where

$$\pi(\boldsymbol{s})[m, :] = \text{softmax}[((n+1) - 2m)\boldsymbol{s} - A_{\boldsymbol{s}}\mathbb{1})/\tau], \tag{1}$$

where $A_{\boldsymbol{s}}$ is the matrix of absolute pairwise differences, i.e., $A_{\boldsymbol{s}}[m, n] = |\boldsymbol{s}_m - \boldsymbol{s}_n|$, $\mathbb{1}$ the column vector of all ones, and $\tau$ a temperature parameter. While Grover et al. (2019) introduce the ranking operator $\pi(\boldsymbol{s})$, this operator does not inherently provide a mechanism for discrete subset selection. Learning $\pi(\boldsymbol{s})$ combines the Gumbel-Softmax trick (Jang et al., 2016; Maddison et al., 2016) with the Plackett-Luce model (PL, Plackett, 1975; Luce et al., 1959).

The PL distribution is a probabilistic model for ranking where items (in our case genes) are selected based on their scores. Given a set of scores $\mathbf{s}$, the probability of a ranking $\pi(\boldsymbol{s})$ follows:

$$P(\pi(\boldsymbol{s})) = \prod_{m=1}^{d} \frac{\pi(\boldsymbol{s})_m}{\sum_{j \in S_m} \pi(\boldsymbol{s})_j}, \tag{2}$$

where $S_m$ represents the set of remaining genes at ranking step $m$. This ranking process ensures that genes with higher scores are more likely to be selected while preserving stochasticity.

The computation of $\pi(\boldsymbol{s})$ involves a temperature parameter $\tau$, which decays over training time, adjusting the ranking sharpness. A decreasing temperature results in more deterministic selection as in the limit of $\tau \to 0$ the softmax function becomes equal to argmax. We follow the exponential annealing schedule suggested in Jang et al. (2016).

However, for $\tau > 0$, the output of the selection of the subset is not a binary mask, and information from all input genes will be used in successive blocks of the pipeline. Using the *straight-through* trick (Bengio et al., 2013), we ensure that YOTO only uses information coming from selected genes by using a binary mask in the forward pass of the model and having the relaxed version of the permutation matrix $\pi(\boldsymbol{s})$ only during the backward pass.

We construct a binary mask $\Gamma_k(\boldsymbol{s})$ by taking the sum over the top $k$ rows of $\pi(\boldsymbol{s})$

$$\Gamma_k(\boldsymbol{s}) = \sum_{m=1}^{k} \pi(\boldsymbol{s})[m,:] \tag{3}$$

At each forward pass, the differentiable top-k selection mask $\Gamma_k(\boldsymbol{s})$ is applied to retain the most informative genes, where the number of selected genes, $k$, is also progressively annealed throughout training to increase sparsity until the final subset size $k$ is reached. Initially, all genes are considered, and over time, YOTO gradually filters out less important ones, retaining only the most relevant ones. This approach ensures that the ranking process remains differentiable, allowing smooth gradient updates while maintaining interpretable feature selection.

## 3.2 Multi-Task Learning

YOTO employs multi-task learning to leverage different (potentially incomplete) sets of supervision labels. Instead of training separate instances of the model for each task, we use a shared feature extractor followed by task-specific prediction heads. Each task module processes its respective output, optimizing for classification or regression, depending on the available labels.

Given a dataset and a set of tasks $\mathcal{T} = \{\mathcal{T}_1, \ldots, \mathcal{T}_T\}$, YOTO optimizes for all tasks jointly:

$$\mathcal{L} = \frac{1}{T} \sum_{t=1}^{T} \mathcal{L}_t, \tag{4}$$

where $\mathcal{L}_t$ is the task-specific loss for task $\mathcal{T}_t$. We define the task-specific loss as

$$\mathcal{L}_t = \frac{1}{N} \sum_{i=1}^{N} l_t(y_t^{(i)}, \hat{y}_t^{(i)}) \quad \text{with} \quad \hat{y}_t^{(i)} = H_t(\boldsymbol{z}^{(i)}), \tag{5}$$

where $\hat{y}_t^{(i)}$ is the prediction of the model for task $\mathcal{T}_t$ and sample $i$, $H_t$ the task-specific prediction network, and $l_t$ the loss function for task $\mathcal{T}_t$. We use cross-entropy loss for classification and mean squared error loss for regression tasks.

$\boldsymbol{z}^{(i)}$ is the output of the shared encoder $E$:

$$\boldsymbol{z}^{(i)} = E(S_k^{(i)}) \quad \text{with} \quad S_k^{(i)} = X^{(i)} \odot \Gamma_k(\boldsymbol{s}) \tag{6}$$

where $\odot$ describes the element-wise operation, $\Gamma_k(\boldsymbol{s})$ is the binary selection mask, and $S_k^{(i)}$ is the selected subset of gene values.

Multi-task learning enables YOTO to leverage shared structure across related tasks, leading to better generalization and more robust feature selection. In addition, tasks with missing labels can still contribute indirectly by helping to refine the shared representation space through supervision of a subset of labeled samples. An important assumption underlying this formulation is that the tasks share underlying biological structure, making the learning of a shared gene subset meaningful.

# 4 Experiments and Results

To validate the efficacy and versatility of YOTO, we conduct a comprehensive set of experiments designed to answer three key questions: (i) Can our end-to-end approach identify more informative gene sets than traditional multi-stage pipelines? (ii) Can YOTO learn to jointly predict multiple tasks rather than requiring one model per label? (iii) Does YOTO generalize across different biological contexts? To address these questions, we evaluate YOTO on two distinct and challenging tasks: first, identifying disease-relevant biomarkers in a single-cell RNA sequencing dataset from COVID-19 patients (i.e., COVID-PBMC (Yao et al., 2021)), and second, selecting compact gene subsets for cell type classification in a spatial transcriptomics dataset (i.e., VISp (Ramsköld et al., 2012)). To assess our model's robustness across different experimental settings, we test a range of gene subset sizes, allowing us to evaluate performance under varying levels of information compression. While YOTO is trained with a multi-task objective in some experiments, all reported metrics correspond to task-specific predictions without any aggregation across tasks. We benchmark our model against a suite of widely-used feature selection methods and the current state-of-the-art, demonstrating superior performance as well as stability across different metrics, in all settings. Implementation details can be found in Appendix A.3.

## 4.1 Datasets and Preprocessing

**COVID-PBMC dataset** To evaluate YOTO's ability to identify disease-relevant biomarkers, we use a public single-cell RNA sequencing (scRNA-seq) dataset of Peripheral Blood Mononuclear Cells (PBMCs) from a cohort of 20 individuals with varying COVID-19 severity (Yao et al., 2021). We refer to this dataset as the COVID-PBMC dataset. The cohort includes healthy controls, patients with moderate symptoms, and patients with severe Acute Respiratory Distress Syndrome (ARDS), comprising approximately 64,000 high-quality cells after preprocessing. For this dataset, we train YOTO with three supervision signals simultaneously: fine-grained cell types (`celltype5`), patient disease status (`group`), and patient ID (`patient`). Dataset splitting is performed at the cell level, with patients and groups overlapping across training, validation, and test sets to ensure that all supervised tasks remain well-defined at evaluation time. Following prior work (Heydari & Oscar, 2022), we remove cells with fewer than 200 expressed genes, genes expressed in fewer than three cells, and cells with more than 10% mitochondrial reads. We also discard cell types with fewer than 100 cells, resulting in a total of nine annotated cell types. As input to all models, we use the top 5,000 highly variable genes (HVGs) identified by Seurat v3 (Stuart et al., 2019).

**VISp dataset** To demonstrate YOTO's utility in the context of spatial transcriptomics, we use a dataset of 22,160 neural cells from the mouse primary visual cortex (VISp) (Ramsköld et al., 2012). A primary challenge in spatial genomics is that imaging-based technologies can only profile a limited number of genes per experiment, making it essential to identify compact and informative gene subsets. We perform multi-task classification on three levels of cell type granularity: `cell_types_25`, `cell_types_50`, and `cell_types_98`. For each task, we filter out classes with fewer than 10 instances, resulting in a total number of 23, 46, and 90 labels, respectively. In line with prior work (Ramsköld et al., 2012), we binarize gene expression values. This preprocessing step reflects the binary nature of many spatial transcriptomics readouts (e.g., fluorescence-based methods), where the key measurement is whether a gene is detected at a location rather than its exact expression magnitude. As input to all models, we use the top 10,000 highly variable genes (HVGs).

### 4.2 Baselines

We compare our method against a diverse set of feature selection approaches that represent complementary paradigms in omics analysis, as summarized in Table 1. Seurat (Satija et al., 2015) and its successor Seurat v3 (Stuart et al., 2019) are included as widely used baselines in single-cell genomics. They perform unsupervised feature selection and serve as representative methods for identifying marker genes without supervision. To assess performance relative to classical supervised feature selection, we further consider two variants of the minimum Redundancy Maximum Relevance (mRMR) criterion (Ding & Peng, 2003): (a) *mRMR-f*, which captures linear dependencies between the features and the target using the F-statistic (derived from ANOVA for discrete targets or from correlation for continuous targets), and (b) *mRMR-rf*, which relies on Random Forest feature importances to capture nonlinear dependencies. Both variants quantify feature redundancy using Pearson correlation. We also include the block HSIC Lasso (Climente-González et al., 2019), a computationally efficient convex kernel-based method derived from HSIC Lasso (Yamada et al., 2014), using $B = 5$ blocks. Finally, we evaluate against the PERSIST model (Covert et al., 2023), a gradient–based approach that integrates feature selection and prediction within an end-to-end architecture, providing a natural point of comparison for our proposed approach.

For classical baseline methods (Seurat, Seurat v3, mRMR-f, mRMR-rf, and Block HSIC Lasso), gene selection and classification are performed as two separate stages. These methods do not produce a predictive model during the selection process; they output only a ranked list of genes. To evaluate the selected subsets, we therefore train an additional downstream classifier, specifically a Random Forest, using only the selected genes. This ensures that all baselines are assessed under the same classification setup and that comparisons are based solely on the quality of the selected gene subsets.

PERSIST differs in that it contains a built-in classifier, but its predictions during training rely on a temperature-controlled masking of the input. Even at a low temperature, the mask is not fully sparse: non-selected genes still contribute small but nonzero signals. This means that PERSIST's internal classifier is evaluated using more information than the final sparsely selected subset actually contains, giving it an inherent advantage that other methods do not have. Despite this, we chose to report PERSIST's internal classification performance at the end of training (temperature 0.01). While this evaluation setup gives PERSIST an advantage, since its not fully sparse mask still allows information from all genes to influence predictions, it enables a direct comparison between the only two single-stage architectures designed to jointly perform gene selection and classification. This allows us to meaningfully assess how YOTO's fully discrete, end-to-end selection mechanism compares to PERSIST's non-fully sparse temperature-based approach under their intended operating conditions.

### 4.3 Experimental Setup

For all experiments, we use a standard 80-20 train-test split and report the mean and standard deviation of performance across three random seeds (i.e., 0, 1, 2). Both gradient-based models, namely YOTO and PERSIST, are trained for 1000 epochs. We report the test performance at the final training epoch. We evaluate performance using macro-averaged F1-score, accuracy, AUROC, and AUPRC. Given that cell type labels in both datasets are highly imbalanced, we prioritize F1-score, AUPRC, and AUROC as they provide a more robust and informative assessment of model performance than accuracy alone. Finally, on the VISp dataset, we evaluate gene subsets of size 16, 32, 64, 128, and 256, while on the COVID-19 dataset, we evaluate subset sizes of 16, 32, and 64, chosen to cover a broad range from highly constrained to information-rich scenarios and to align with common experimental constraints.

In addition to reporting the results of YOTO in its fully integrated setting, we also report results obtained by coupling the gene subsets selected by YOTO with a downstream Random Forest classifier, denoted as YOTO (FS only). This two-stage evaluation facilitates a more direct comparison with classical feature selection baselines by isolating the quality of the selected gene panels from the effects of joint training.

## 4.4 Results

Our comprehensive evaluation empirically demonstrates the strong performance of the proposed end-to-end multi-task method across a wide range of settings. In particular, we highlight three key advantages of YOTO: (i) its consistent efficacy across different numbers of selected genes (Section 4.4.1), (ii) its ability to perform multi-task learning competitively *without requiring additional training steps or task-specific retraining* (Section 4.4.2), and (iii) its robustness across multiple evaluation metrics, including those designed for imbalanced datasets (Section 4.4.3). Furthermore, an ablation study on the single-task setting (Section 4.4.4) shows that YOTO's advantage over the strongest deep learning baseline is not due to model size (chosen to support multi-task learning) but stems from its sparse and fully differentiable selection mechanism, which allows it to learn discrete, informative gene subsets directly during training.

### 4.4.1 Consistent Performance Across Different Gene Panel Sizes

**Experiment** We evaluate YOTO across a range of gene subset sizes, an essential consideration for practical applications (e.g., spatial transcriptomics or targeted profiling) where only a limited number of genes can be measured (Covert et al., 2023). On the VISp dataset, we assess subsets of 16, 32, 64, 128, and 256 genes; on the COVID-PBMC dataset, we consider subsets of 16, 32, and 64 genes.

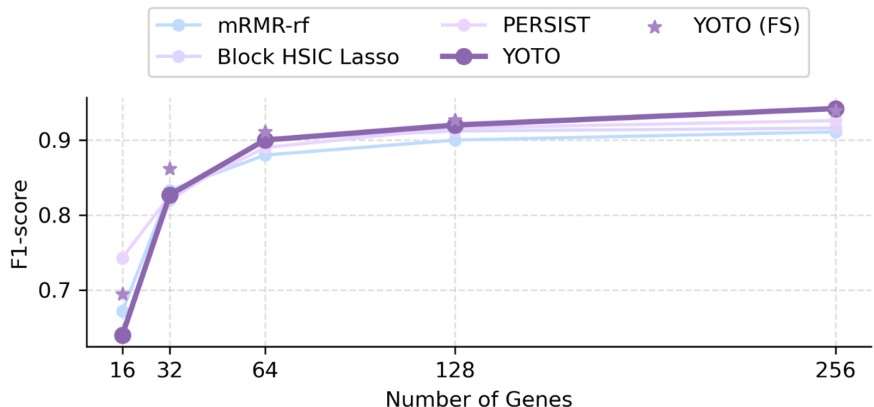

Figure 2: **Downstream F1-score across different number of selected genes.** F1-scores for the `cell_types_25` task on the VISp dataset using gene subsets of sizes $k \in \{16, 32, 64, 128, 256\}$. The Seurat and mRMR-f baselines are omitted due to consistently low performance.

**Results** As shown in Figure 2 and Table 6 YOTO consistently demonstrates state-of-the-art F1-scores across nearly all subset sizes on the VISp dataset. The only exception occurs at the smallest subset size ($k = 16$), where performance is highly sensitive to individual gene choice, a known challenge when the number of input features is extremely constrained. For all larger panel sizes ($\geq 32$ genes), YOTO consistently outperforms both classical and gradient-based baselines and exhibits stronger improvement as the panel size increases. On the COVID-PBMC dataset (Table 7), we observe a similar trend, but with one notable exception: the classical information-theoretic baseline mRMR-f achieves the highest F1-scores at 16 and 32 genes, while YOTO ranks second and becomes the best-performing method at 64 genes. This behavior reflects the dataset and chosen task: the coarse-grained COVID-PBMC labels are strongly associated with a few highly predictive genes, which allows simple relevance-based criteria like mRMR-f to excel when the panel is extremely small.

**Discussion** The strengthening performance of YOTO as the panel size increases suggests that it is not merely selecting individually predictive genes but is also capturing interactions and complementary gene sets often overlooked by simpler selection methods. This behavior is especially advantageous in single-cell settings, where cell identity and disease signatures often arise from combined gene sets rather than isolated markers. This trend is consistent with YOTO's discrete, end-to-end formulation: as the subset size increases,

the model can better exploit coordinated biological signals through the shared representation learned jointly with the prediction objective. In contrast, at very small panel sizes the selection problem becomes highly constrained, making performance more sensitive to individual gene choices. Importantly, when evaluating the selected gene panels in isolation using a random forest classifier (YOTO FS), YOTO is superior or hihgly competitive across most subset sizes (see Tables 8 and 9). The only exception at $k = 16$ can be attributed to PERSIST, which relies on a soft masking mechanism that allows non-selected genes to continue contributing to prediction, an advantage that does not reflect the true predictive capacity of the selected subset itself. YOTO's competitive performance even at very small subset sizes, combined with its clear advantage once moderate panel sizes are allowed, highlights its robustness across the number of selected genes, datasets, and biological contexts. Importantly, YOTO provides these benefits within an end-to-end, single-model training setup, unlike classical baselines that require retraining separate classifiers for evaluation. These properties make YOTO well-suited for both cost-sensitive experimental designs and for high-resolution studies requiring richer gene subsets.

### 4.4.2 Multi-Task Learning

**Experiment**  We evaluate YOTO's ability to learn from multiple supervisory signals within a single end-to-end model. Instead of training separate models for each task, YOTO jointly uses all available labels during training. On the COVID-PBMC dataset, a single model simultaneously predicts fine-grained cell types (`celltype5`), patient disease status (`group`), and patient identity (`patient`). Likewise, on the VISp dataset, one model jointly predicts three levels of cell-type granularity (`cell_types_25`, `cell_types_50`, `cell_types_98`). This multi-task formulation allows the model to exploit shared structure across tasks.

Table 2: **Multi-task performance on the COVID-PBMC dataset.** F1-scores (%) for three prediction tasks learned jointly within a single model (using 64 selected genes). "#models" indicates the number of models required by each method to handle feature selection and prediction for all tasks. In the Pipeline column, "FS → RF" denotes methods that perform feature selection (FS) independently and are evaluated using a downstream Random Forest (RF) trained on the selected genes; "Integrated" denotes methods that couple gene selection and prediction within a single model. The best performing model for each task is highlighted in bold, while the second best one is underlined.

| Pipeline | Method | #models | celltype5 | group | patient |
|---|---|---|---|---|---|
| FS → RF | Seurat | 4 | $38.5 \pm 1.5$ | $35.5 \pm 0.1$ | $10.1 \pm 0.5$ |
| | Seurat v3 | 4 | $54.5 \pm 0.9$ | $53.9 \pm 0.4$ | $29.1 \pm 0.6$ |
| | mRMR-f | 6 | $84.8 \pm 0.5$ | $85.4 \pm 0.2$ | $70.1 \pm 0.2$ |
| | mRMR-rf | 6 | $78.3 \pm 0.9$ | $\underline{93.1 \pm 0.2}$ | $71.7 \pm 0.4$ |
| | Block HSIC Lasso | 6 | $79.6 \pm 0.2$ | $91.6 \pm 0.2$ | $69.8 \pm 0.6$ |
| | **YOTO (FS only)** | 4 | $\underline{85.3 \pm 0.9}$ | $92.8 \pm 0.2$ | $\underline{72.7 \pm 0.1}$ |
| Integrated | PERSIST | 3 | $76.5 \pm 0.7$ | $87.1 \pm 0.6$ | $58.8 \pm 1.0$ |
| | **YOTO** | **1** | $\mathbf{86.0 \pm 0.4}$ | $\mathbf{94.5 \pm 0.1}$ | $\mathbf{74.9 \pm 0.2}$ |

**Results**  Table 2 and Table 3 show the F1-scores across all tasks using 64 selected genes. YOTO, which is trained only once in a joint multi-task setting, matches or surpasses the performance of the other methods, which in contrast require separate training runs for each task. The greatest improvements are observed on the more challenging tasks (e.g., `patient` and `cell_types_50`), suggesting that learning from multiple biological signals improves generalization and yields more informative gene representations. A notable practical advantage of YOTO is model efficiency: it requires only one training instance per dataset as it jointly performs feature selection and prediction for all tasks. In contrast, baselines require between 3 and 6 separate models, depending on whether their feature selection and classification must be trained independently for each task (as in mRMR-f, mRMR-rf, and Block HSIC Lasso) or are only partially shared (as in Seurat and Seurat v3, where subset selection is performed once independent of the task). PERSIST also integrates selection and prediction but supports only a single task per model and therefore must be retrained for each task.

Table 3: **Multi-task performance on the VISp dataset.** F1-scores (%) for three cell-type classification tasks learned jointly within a single model (using 64 selected genes). The column "#models" reports the number of models required by each method to perform feature selection and prediction across all tasks.

| Pipeline | Method | #models | cell_types_25 | cell_types_50 | cell_types_98 |
|---|---|---|---|---|---|
| FS $\rightarrow$ RF | Seurat | 4 | $66.2 \pm 2.1$ | $50.3 \pm 1.7$ | $39.5 \pm 0.8$ |
| | Seurat v3 | 4 | $49.2 \pm 1.0$ | $36.5 \pm 1.7$ | $25.5 \pm 1.7$ |
| | mRMR-f | 6 | $81.1 \pm 2.0$ | $64.8 \pm 1.5$ | $50.1 \pm 1.8$ |
| | mRMR-rf | 6 | $88.0 \pm 1.6$ | $75.7 \pm 1.1$ | $65.0 \pm 1.5$ |
| | Block HSIC Lasso | 6 | $90.1 \pm 1.0$ | $78.7 \pm 1.1$ | $\underline{69.7 \pm 0.6}$ |
| | **YOTO (FS only)** | 4 | $\mathbf{91.1 \pm 0.8}$ | $\mathbf{82.6 \pm 0.7}$ | $\mathbf{72.1 \pm 0.5}$ |
| Integrated | PERSIST | 3 | $89.0 \pm 1.4$ | $74.4 \pm 0.9$ | $60.3 \pm 1.3$ |
| | **YOTO** | 1 | $\underline{90.6 \pm 0.9}$ | $\underline{79.9 \pm 1.9}$ | $66.8 \pm 1.9$ |

Additionally, when training task-specific downstream classifiers on the gene subsets selected by the multi-task YOTO model, these classifiers consistently outperform all baselines across subtasks and, in some cases, exceed the performance of end-to-end YOTO on the COVID-PBMC dataset. This highlights the intrinsic quality and cross-task relevance of the selected gene panels.

**Discussion** These results show that multi-task learning enables YOTO to leverage complementary biological information (e.g., fine-grained cell types, disease labels, and patient-level variation) within a unified representation. This shared structure leads to stronger performance on difficult tasks and avoids redundant training across subtasks. Unlike baselines that must perform feature selection and classification separately for each label, YOTO jointly learns a shared gene subset and task-specific predictors in a single end-to-end framework.

Note that our setting is inherently a cell-level prediction problem, where both supervision (e.g., cell type) and evaluation are defined per cell. A cell-level split is therefore appropriate, as it assesses generalization to unseen cells rather than unseen patients, which is the standard formulation in single-cell representation learning. Within this framework, patient identity and disease group represent structured sources of variation expressed at the cell level, and can be used as auxiliary supervision to guide representation learning. At the same time, we emphasize that patient prediction is not a biological objective per se, but is included as an auxiliary task to model inter-patient variability and batch-like effects, which are known confounders. In highly constrained regimes (for example, very small gene subset sizes), jointly optimizing multiple tasks can introduce trade-offs, where patient-informative genes may be retained at the expense of optimal cell-type markers. This effect diminishes as the subset size increases and reflects a limitation of extreme sparsity rather than a failure of the approach. We acknowledge that cell-level splits provides a weaker notion of generalization than patient-disjoint evaluation and may allow models to leverage patient-specific signals, particularly under strong sparsity constraints. This highlights a trade-off between optimizing cell-level predictive performance and identifying biomarkers that generalize across patients, which we leave to future work.

Overall, multi-task learning enhances YOTO's capacity to identify gene subsets that are informative across diverse biological signals, provided that the tasks share underlying biological structure. When this assumption holds, learning a shared subset can improve robustness and interpretability. However, when it does not, task-specific selection may be more appropriate.

### 4.4.3 Robustness Across Multiple Evaluation Metrics

**Experiment** We assess YOTO's robustness across several evaluation metrics, including those designed to handle the substantial class imbalance present in both datasets. Statistics on the datasets class distribution are provided in Section A.1. Given the skewed distribution of cell types (particularly in the VISp dataset) we prioritize metrics that more accurately reflect performance in imbalanced settings. Accordingly, we evaluate models using F1-score, the Area Under the Precision–Recall Curve (AUPRC), and the Area Under the

Receiver Operating Characteristic Curve (AUROC), in addition to accuracy. We report the average and standard deviations of each metric over three seeds.

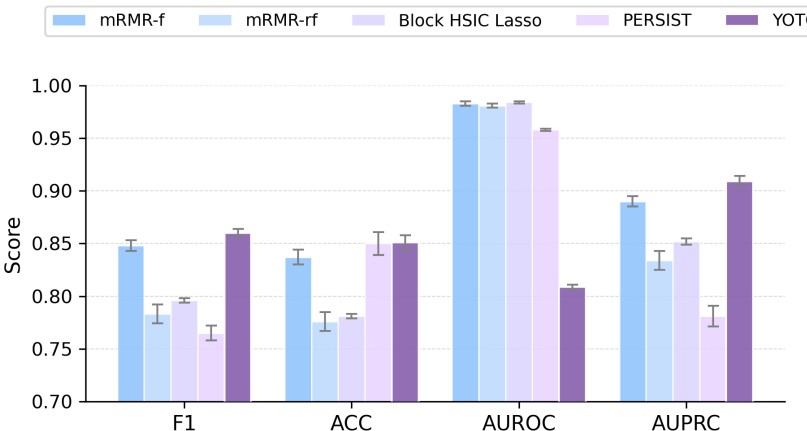

Figure 3: **Comprehensive evaluation using multiple classification metrics.** F1-score, Accuracy, AUROC, AUPRC) for the `celltype5` task on the COVID-PBMC dataset using a gene subset of size $k = 64$.

**Results** Figure 3 shows the results for the `celltype5` task on the COVID-PBMC dataset using 64 genes. YOTO consistently outperforms all baselines in terms of F1-score and AUPRC. It matches the strongest baseline with respect to accuracy, but achieves a slightly lower AUROC, reflecting differences in how these metrics capture model behavior.

**Discussion** This difference in AUROC is expected. AUROC measures global ranking quality across all possible classification thresholds, whereas F1-score and AUPRC emphasize model performance near the threshold region most relevant for imbalanced classification. In highly skewed settings like the cell type classification, AUPRC and F1-score provide a more informative picture of the model's ability to correctly identify minority cell types. YOTO's superior results on these metrics indicate that it makes more reliable positive predictions and better captures rare cell populations, even if its score ranking across the full threshold range is less separated than that of some baselines.

Taken together, these findings highlight the robustness of YOTO across evaluation metrics and underscore its practical value for accurate detection of underrepresented cell types in real-world single-cell analyses.

### 4.4.4 Effect of the Subset Selection Module on Performance

**Experiment** To isolate the effect of YOTO's subset selection module, we conduct an ablation study in which YOTO is trained with the same architecture, temperature parameters, and single-task objective as PERSIST. This alignment ensures that any observed performance differences arise solely from the feature selection mechanism rather than differences in model capacity, multi-task supervision, or optimization setup.

**Results** Table 4 reports F1-scores on the VISp dataset under identical conditions. YOTO matches or outperforms PERSIST for all but the most constrained subset size ($k = 16$). For gene subset sizes ranging from 32 to 256 genes, YOTO achieves consistently higher F1-scores, confirming that its discrete selection mechanism is effective even without multi-task learning or larger architectures. In the extreme setting of 16 genes, however, PERSIST shows a notable performance advantage.

**Discussion** These results show that YOTO's improvements in the main experiments are not solely attributable to architectural size or the benefits of multi-task training. Instead, they arise from the feature selection mechanism itself. The underperformance at $k = 16$ reflects the inherent difficulty of selecting highly informative genes under such tight constraints; notably, YOTO's performance at the smallest subset

Table 4: **Ablation Study on the Subset Selection Module.** Performance comparison between YOTO and PERSIST on the `cell_types_25` (VISp), using identical architecture, single-task training, and temperature parameters. This ablation isolates the effect of the subset selection mechanism by controlling for all other architectural and optimization factors. Average F1-scores (%) over three seeds are reported.

| # Genes | PERSIST | YOTO |
|---|---|---|
| 256 | $92.6 \pm 1.1$ | $\mathbf{93.9 \pm 1.2}$ |
| 128 | $91.6 \pm 2.0$ | $\mathbf{92.7 \pm 1.0}$ |
| 64 | $89.0 \pm 1.4$ | $\mathbf{92.1 \pm 1.9}$ |
| 32 | $82.6 \pm 4.1$ | $\mathbf{82.9 \pm 2.9}$ |
| 16 | $\mathbf{74.3 \pm 0.8}$ | $55.2 \pm 4.1$ |

size improves substantially in the multi-task setting, indicating that the limitation is not fundamental to the method. However, the apparent advantage of PERSIST at $k = 16$ should be interpreted with caution: because PERSIST relies on a non-sparse temperature-masked input, all genes (and not just the selected subset) can still influence the prediction. This gives PERSIST an inherent advantage in the small-subset regime. Importantly, this advantage would not hold in practice, since downstream use of PERSIST's selected genes requires a fully sparse input, which would yield lower performance. We nevertheless allow PERSIST this favorable (and practically unrealistic) setting here, as the goal of the ablation is to isolate and compare the behavior of the selection mechanisms themselves under identical architectural and training conditions.

Overall, this ablation confirms that YOTO's gains stem from its sparse, end-to-end selection module rather than architectural differences or the multi-task training. It further highlights that our approach is especially well-suited for realistic gene subset sizes, where leveraging interactions among selected genes becomes increasingly advantageous.

### 4.4.5 Isolating the Multi-Task Setting

**Experiment** To isolate the effect of multi-task learning (MTL) and enable a direct comparison with the single-task learning (STL) setting, we reproduce the experiment from Section 4.4.4 using a multi-task objective. In this setting, tasks `cell_types_50` and `cell_types_98` are included as auxiliary supervision signals, each with its own task-specific head, alongside the primary task `cell_types_25`, on which performance is reported.

**Results** Results are shown in Table 5. The first two columns use the exact same architecture (two hidden layers), differing only in the training paradigm (STL vs. MTL). The final column reports results obtained with a slightly larger architecture (one additional hidden layer) for the MTL setting, which we found helps stabilize optimization for the more complex multi-task objective. We highlight in bold the best performance between STL and MTL when using identical architectures.

Overall, when using the same architecture, YOTO achieves higher performance in the STL setting than in the MTL setting (columns 1 and 2), with MTL exhibiting larger variance across seeds. With the addition of a single extra hidden layer, MTL performance becomes comparable to or exceeds STL (column 3).

**Discussion** This behavior is consistent with prior work (Kurin et al., 2022) and suggests that jointly learning multiple tasks increases optimization difficulty when model capacity is limited, as also reflected by the higher variance observed in the MTL setting. The improved performance obtained with a slightly larger architecture indicates that additional capacity can help mitigate this effect and better exploit multi-task supervision. Overall, while multi-task learning represents an added flexibility of the YOTO framework and can be beneficial in certain regimes, particularly under the most constrained panel size ($k = 16$), it is neither strictly necessary nor always optimal for this task.

### 4.4.6 Robustness of Selected Genes Across Seeds

**Experiment** To assess the robustness of gene subset selection across different model initializations and dataset splits, we evaluate the stability of YOTO's selected genes across different random seeds. In this

Table 5: **Ablation Study of the Multi-task Setting.** Performance comparison of YOTO in single-task (STL) and multi-task (MTL) settings on the `cell_types_25` task (VISp). The first two columns use identical architectures (two hidden layers), isolating the effect of STL vs. MTL, while the final column uses a slightly larger architecture (one additional hidden layer) to stabilize optimization in the MTL setting. This ablation controls for architectural and optimization factors and complements the results in Figure 2, which also rely on increased capacity for MTL. Average F1-scores (%) over three seeds are reported, and the best result between STL and MTL with identical architectures is highlighted in bold.

| # Genes | YOTO (STL) | YOTO (MTL) | *YOTO (MTL, larger)* |
|---|---|---|---|
| 256 | **93.9 ± 1.2** | 93.7 ± 1.4 | *94.2 ± 0.9* |
| 128 | **92.7 ± 1.0** | 90.8 ± 1.9 | *92.0 ± 0.7* |
| 64 | **92.1 ± 1.9** | 89.4 ± 1.5 | *90.0 ± 2.2* |
| 32 | **82.9 ± 2.9** | 81.8 ± 4.6 | *82.7 ± 2.7* |
| 16 | 55.2 ± 4.1 | **56.8 ± 7.0** | *64.0 ± 4.0* |

experiment, we focus on the model selecting $k = 64$ genes for the VISp dataset, for which multi-task classification performance is reported in Table 3. This configuration achieves competitive performance across all tasks. Because feature selection in high-dimensional transcriptomic data is inherently stochastic and subject to strong gene-gene correlations, robustness is evaluated at multiple levels, including [...]. To assess the robustness of the genes selected by YOTO across seeds, we look at the genes selected by YOTO whose classification results are reported in Table 3 (multi-task performance of YOTO on the VISp dataset for a subset of size $k = 64$) as this model shows competitive performance across all tasks. Implementation details can be found in Section A.3.

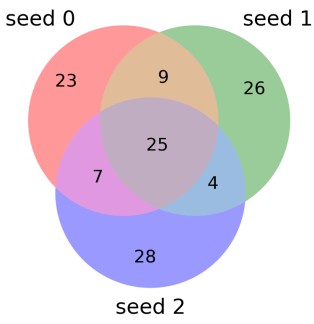

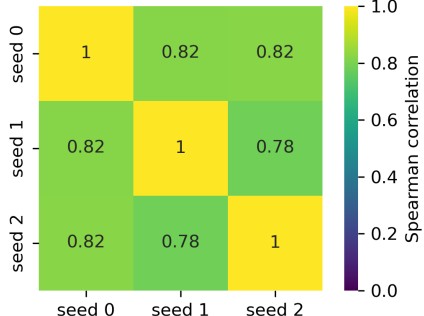

(a) Gene-level overlap across seeds.

(b) GO term rank concordance across seeds.

Figure 4: **Robustness of gene selection across random seeds.** Despite modest gene-level overlap, functional enrichment is highly consistent across seeds. (Left) Venn diagram showing overlap of genes selected by YOTO for multi-task training on the VISp dataset using a subset size of $k = 64$. (Right) Pairwise Spearman correlation of ranked Gene Ontology (GO) biological process terms across seeds, indicating strong functional concordance.

**Results** Figure 4 (left) in the Appendix shows the overlap of selected genes across three independent random seeds. The average pairwise Jaccard Index is $0.329 \pm 0.028$, indicating that approximately 33% of genes were shared between runs. Given that the task involves selecting 64 genes from a pool of 10000 candidates, this overlap is more than three orders of magnitude higher than expected under random selection. In addition, we assess whether known marker genes (Tasic et al., 2018) were preferentially selected and stably recovered across seeds. Marker genes were significantly enriched among genes selected by all three runs (Fisher's exact test, odds ratio $\approx 12$, $p < 10^{-3}$), indicating that biologically established markers were consistently prioritized by the model despite stochastic variability in gene identity. To assess robustness at the functional level, we compared Gene Ontology (GO) biological process enrichment across seeds. Despite modest gene-level overlap, enriched GO terms were highly consistent across runs. Figure 4 (right) show the

Spearman correlation of the top 30 GO terms processes across seeds. Shared processes included synaptic transmission, axon guidance, cell-cell adhesion, and calcium-dependent signaling, indicating that stochastic variation in gene identity occurs within stable biological programs rather than reflecting divergent functional solutions.

**Discussion** Although direct gene-level overlap across random seeds is necessarily limited under sparse selection in a high-dimensional and highly correlated feature space, the observed overlap is substantially higher than expected by chance and is enriched for biologically established marker genes. This indicates that stochasticity in gene identity reflects redundancy among correlated features rather than instability of the learned representations. Importantly, the subset of genes consistently selected across all seeds includes genes that are part of functionally coherent genes networks implicated in neuronal identity, connectivity, and signaling. They are involved in synaptic transmission and glutamatergic signaling (e.g., *GRIN3A*, *GRIK1*, *NPTX2*), axon guidance and cell–cell adhesion (*CDH9*, *CDH13*, *PCDH8*, *CNTNAP5A*, *SEMA3E*), and intracellular signaling and transcriptional regulation (*PDE1A*, *NR2F2*, *NPAS1*). Several of these genes are also present in known marker lists (Tasic et al., 2018), while others capture aspects of neuronal function that are not strictly canonical but are nevertheless highly informative in the multi-task prediction setting.

At the same time, many genes are selected in a seed-specific manner, suggesting the existence of multiple near-equivalent gene subsets that support comparable predictive performance. This behavior is consistent with the strong functional redundancy characteristic of transcriptomic data, where correlated genes can act as interchangeable proxies for shared biological processes. Together with the observed consistency of enriched Gene Ontology biological processes across seeds, these results indicate that YOTO reliably converges on stable functional programs even when individual gene identities vary. This motivates a complementary analysis of robustness across gene subset sizes, where we assess whether increasing panel size leads to progressive stabilization of a core gene set or continued reshuffling, thereby providing further insight into the structure and robustness of the learned gene representations.

### 4.4.7 Robustness of Selected Genes Across Gene Subset Sizes

**Experiment** We assess the robustness of gene selection across different subset sizes by quantifying the degree to which gene panels selected at smaller sizes are preserved when the panel size increases. We consider the gene subsets selected by YOTO for the experiments reported in Section 4.4.1. Because direct set similarity measures such as the Jaccard index can be misleading when comparing sets of different cardinalities, we instead quantify nestedness using a directional containment measure equivalent to recall in information retrieval. Specifically, for two panel sizes $k_1 < k_2$, nestedness is defined as

$$\text{Nestedness}(k_1 \rightarrow k_2) = \frac{|S_{k_1} \cap S_{k_2}|}{|S_{k_1}|}, \tag{7}$$

which measures the fraction of genes selected at size $k_1$ that are retained in the larger panel $S_{k_2}$.

**Results** Results are shown in Figure 5. Nestedness increases monotonically with gene subset size, accompanied by a reduction in variability across random seeds. In particular, transitions involving larger panels exhibit higher and more consistent containment, indicating that genes selected at intermediate and large subset sizes are increasingly preserved under further expansion. This trend suggests that YOTO progressively converges on a stable core of genes as model capacity increases.

**Discussion** These results indicate that although gene selection at small subset sizes is sensitive to correlated feature substitution, YOTO increasingly converges on a robust set of core genes as the subset size grows. Importantly, YOTO re-optimizes gene selection independently at each panel size, allowing genes to be replaced when more informative or complementary alternatives become available. The observed increase in nestedness therefore reflects adaptive stabilization rather than a fixed ranking of genes, highlighting YOTO's ability to balance flexibility at small scales with robustness at larger gene panel sizes.

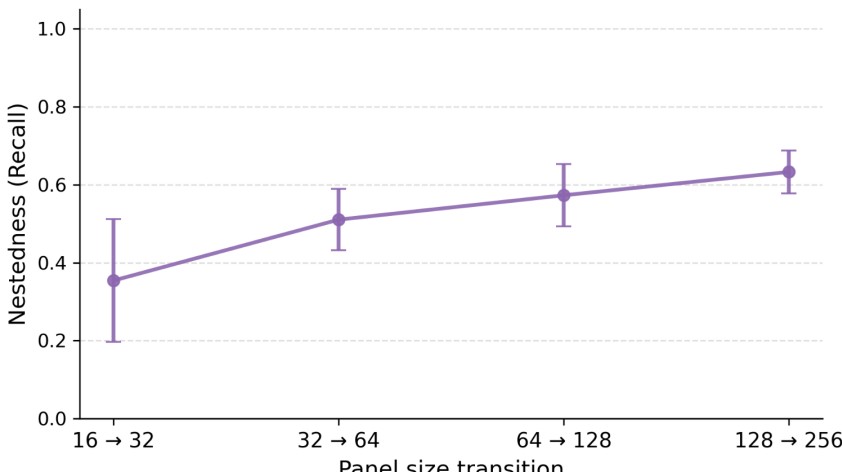

Figure 5: **Nestedness of selected gene subsets across sizes.** Directional containment (recall) quantifies the fraction of genes selected at a smaller panel size that are retained at the next larger size. Error bars denote standard deviation across random seeds. Gene subsets correspond to those selected by YOTO for multi-task training on the VISp dataset.

## 5 Conclusion

In this work, we presented YOTO, a gradient-based learning framework for end-to-end gene selection and multi-task learning in single-cell transcriptomics. Unlike classical feature selection methods or post hoc attribution techniques, YOTO learns to do sparse selection and predict simultaneously, enforcing discrete gene choices during training. This eliminates the need for downstream task retraining, tightly couples feature selection with supervised objectives, and yields interpretable and reproducible gene panels.

A key advantage of YOTO is its ability to operate in a multi-task learning setting, allowing a single model to leverage heterogeneous labels. This reduces the need to train separate models for each task and encourages the discovery of gene subsets that are informative across multiple biological signals. Across both human COVID-PBMC and mouse VISp datasets, YOTO matches or outperforms state-of-the-art baselines, with clear gains at moderate and larger panel sizes while remaining competitive even under very tight gene subset sizes. Our ablation study of the selection module confirms that these improvements stem from YOTO's sparse, fully differentiable selection module, rather than architectural size or multi-task supervision alone. The ability to perform well when selecting small subsets from high-dimensional inputs is particularly promising for applications such as targeted profiling and spatial transcriptomics, where measurement capacity is limited.

More broadly, YOTO provides a general-purpose strategy for discrete feature selection that is not restricted to gene expression. Its design is applicable to other high-dimensional domains (e.g., proteomics, epigenomics, and multi-omics integration) and to non-biological settings where small feature sets are crucial. Together, these properties position YOTO as a practical and versatile solution for discrete feature selection in high-dimensional settings.

Finally, YOTO naturally supports learning from partially labeled datasets, where not every sample is annotated for every task. Prior work has shown that multi-task learning in such settings can outperform single-task learning by leveraging shared representations learned by a common encoder, allowing supervision from one task to benefit others (Liu et al., 2007; He et al., 2020). Importantly, this benefit does not arise from treating missing labels as unsupervised data, but from information sharing across tasks through the shared encoder. While this intuition is well established in the multi-task learning literature, it would deserve to be validated in specific experiments and therefore represents an interesting direction for future work.

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

# A    Appendix

## A.1    Dataset Imbalance

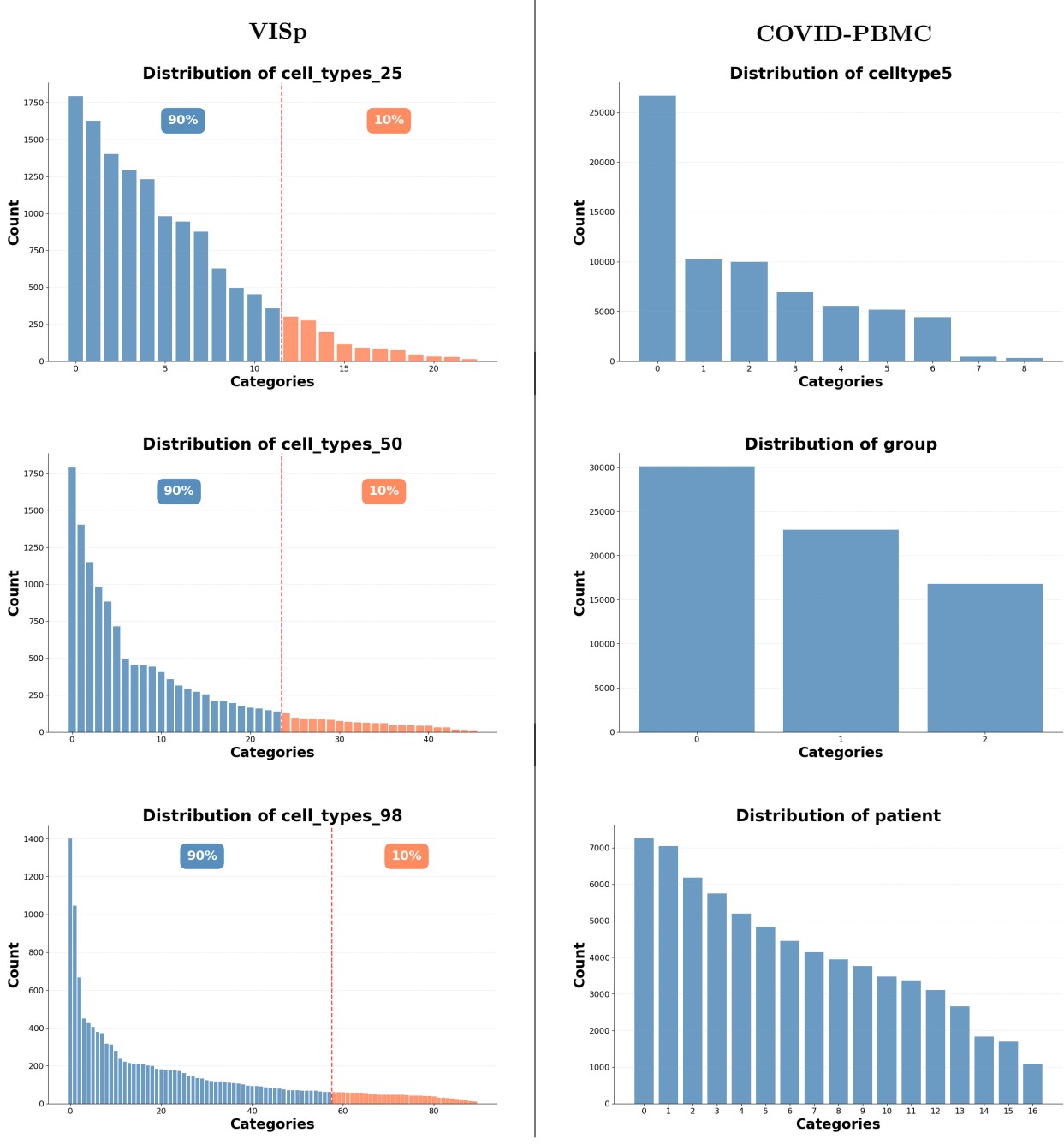

Figure 6: Label distributions for all tasks in the VISp and COVID-PBMC datasets. In the VISp panels, the dashed vertical line, shown only in the VISp distributions, denotes the cumulative threshold dividing the first 90% of observations from the remaining tail.

Figure 6 shows the class distributions of the mouse VISp and COVID-PBMC datasets, ordered from the most to the least frequent class for each subtask. Both datasets exhibit substantial class imbalance. For VISp, the number of label groups increases across tasks; however, in all cases, roughly half of the categories

account for the vast majority of observations. In contrast, the targets in the COVID-PBMC dataset contain fewer categories per task, but still display a noticeable class imbalance.

## A.2 Extended results

To complement the main results presented in Section 4.4, we provide additional quantitative evaluations across multiple panel sizes, datasets, and performance metrics. These extended experiments allow us to assess the robustness of our method under varying levels of feature sparsity and class imbalance, and to compare it more comprehensively against the baselines for both VISp and COVID-PBMC. The following tables summarize these analyses and highlight the consistency of our model's improvements across diverse settings.

Table 6: F1-scores (%) for classifying `cell_types_25` (VISp) using different gene subset sizes. In the Pipeline column, "FS → RF" denotes methods that perform feature selection (FS) independently and are evaluated using a downstream Random Forest (RF) trained on the selected genes; "Integrated" denotes methods that couple gene selection and prediction within a single model.

| Pipeline | Method | Number of Genes | | | | |
|---|---|---|---|---|---|---|
| | | 16 | 32 | 64 | 128 | 256 |
| FS → RF | Seurat | $34.0 \pm 2.8$ | $48.3 \pm 4.1$ | $66.2 \pm 2.1$ | $77.4 \pm 3.0$ | $79.5 \pm 3.8$ |
| | Seurat v3 | $19.4 \pm 2.7$ | $31.4 \pm 1.9$ | $49.2 \pm 1.0$ | $65.5 \pm 1.9$ | $74.9 \pm 2.3$ |
| | mRMR-f | $45.6 \pm 5.4$ | $68.5 \pm 0.3$ | $81.1 \pm 2.0$ | $89.2 \pm 1.9$ | $90.4 \pm 1.3$ |
| | mRMR-rf | $67.2 \pm 1.6$ | $\underline{83.3 \pm 0.9}$ | $88.0 \pm 1.6$ | $90.0 \pm 2.0$ | $91.1 \pm 2.1$ |
| | Block HSIC Lasso | $64.4 \pm 1.1$ | $82.0 \pm 1.0$ | $\underline{90.1 \pm 1.0}$ | $91.2 \pm 0.7$ | $91.6 \pm 2.0$ |
| | **YOTO (FS only)** | $\underline{69.5 \pm 3.2}$ | $\mathbf{86.3 \pm 0.4}$ | $\mathbf{91.1 \pm 0.8}$ | $\mathbf{92.7 \pm 0.8}$ | $\underline{93.9 \pm 1.3}$ |
| Integrated | PERSIST | $\mathbf{74.3 \pm 0.8}$ | $82.6 \pm 4.1$ | $89.0 \pm 1.4$ | $91.6 \pm 2.0$ | $92.6 \pm 1.1$ |
| | **YOTO** | $64.0 \pm 4.0$ | $82.7 \pm 2.7$ | $\underline{90.1 \pm 0.3}$ | $\underline{92.0 \pm 0.7}$ | $\mathbf{94.2 \pm 0.9}$ |

Table 7: F1-scores (%) for classifying `celltype5` (COVID-PBMC) using different gene subset sizes. In the Pipeline column, "FS → RF" denotes methods that perform feature selection (FS) independently and are evaluated using a downstream Random Forest (RF) trained on the selected genes; "Integrated" denotes methods that couple gene selection and prediction within a single model.

| Pipeline | Method | Number of Genes | | |
|---|---|---|---|---|
| | | 16 | 32 | 64 |
| FS → RF | Seurat | $13.4 \pm 3.8$ | $23.4 \pm 3.0$ | $38.5 \pm 1.5$ |
| | Seurat v3 | $33.6 \pm 0.5$ | $37.2 \pm 0.2$ | $54.5 \pm 0.9$ |
| | mRMR-f | $\mathbf{78.0 \pm 0.3}$ | $\mathbf{81.7 \pm 0.3}$ | $84.8 \pm 0.5$ |
| | mRMR-rf | $69.8 \pm 0.4$ | $76.2 \pm 0.8$ | $78.3 \pm 0.9$ |
| | Block HSIC Lasso | $70.9 \pm 1.2$ | $78.3 \pm 0.4$ | $79.6 \pm 0.2$ |
| | **YOTO (FS only)** | $65.1 \pm 1.7$ | $78.0 \pm 0.4$ | $\underline{85.3 \pm 0.9}$ |
| Integrated | PERSIST | $62.8 \pm 3.4$ | $73.7 \pm 1.1$ | $76.5 \pm 0.7$ |
| | **YOTO** | $\underline{71.5 \pm 3.2}$ | $\underline{79.5 \pm 0.9}$ | $\mathbf{86.0 \pm 0.4}$ |

Tables 6 and 7 report the average F1 scores for different panel sizes for each dataset, respectively. For VISp, when predicting `cell_types_25`, our model consistently outperforms the baselines for all panel sizes except the smallest subset ($k = 16$). For the `celltype5` prediction task in COVID-PBMC, our model achieves the best performance for the largest panel size, while ranking second for smaller gene subsets. In both datasets, our model benefits from larger panel sizes, showing steeper performance improvements as the number of selected genes increases, while remaining competitive across a wide range of subset sizes.

Table 8 reports the mean and standard deviation of the F1 score, accuracy, AUROC, and AUPRC on the VISp dataset for classifying `cell_types_25`. For metrics that are more sensitive to class imbalance, such

Table 8: Cell type classification performance (%) using 64 selected genes on the VISp dataset for the `cell_types_25` task. Our single-stage model achieves the best performance across different metrics.

| Pipeline | Method | F1 | ACC | AUROC | AUPRC |
|---|---|---|---|---|---|
| FS → RF | Seurat | 66.2 ± 2.1 | 65.9 ± 2.4 | 98.7 ± 0.1 | 72.7 ± 0.8 |
| | Seurat v3 | 49.2 ± 1.0 | 52.6 ± 2.3 | 95.2 ± 0.6 | 55.7 ± 1.8 |
| | mRMR-f | 81.1 ± 2.0 | 82.7 ± 2.1 | 99.6 ± 0.1 | 87.2 ± 1.2 |
| | mRMR-rf | 88.0 ± 1.6 | 86.8 ± 1.3 | **99.9 ± 0.0** | 92.7 ± 1.0 |
| | Block HSIC Lasso | 90.1 ± 1.0 | 88.3 ± 1.1 | **99.9 ± 0.0** | 93.9 ± 0.9 |
| | **YOTO (FS only)** | **91.1 ± 0.8** | 90.3 ± 1.2 | **99.9 ± 0.0** | 94.2 ± 0.9 |
| Integrated | PERSIST | 89.0 ± 1.4 | **95.6 ± 0.2** | 99.8 ± 0.1 | 92.9 ± 1.2 |
| | **YOTO** | 90.1 ± 0.3 | 88.7 ± 0.2 | 91.9 ± 0.1 | **94.5 ± 1.3** |

as F1 and AUPRC, our method clearly outperforms all other baselines. It remains competitive in terms of accuracy and shows more stable performance across different seeds, particularly for F1 and accuracy.

Table 9: Cell type classification performance (%) using 64 selected genes on the COVID-PBMC dataset for the `celltype5` task. Our single-stage model achieves the best performance across different metrics.

| Pipeline | Method | F1 | ACC | AUROC | AUPRC |
|---|---|---|---|---|---|
| FS → RF | Seurat | 38.5 ± 1.5 | 40.9 ± 1.7 | 74.7 ± 1.8 | 40.4 ± 2.1 |
| | Seurat v3 | 54.5 ± 0.9 | 57.7 ± 0.3 | 88.0 ± 0.1 | 59.3 ± 0.1 |
| | mRMR-f | 84.8 ± 0.5 | 83.7 ± 0.7 | 98.3 ± 0.2 | 89.0 ± 0.5 |
| | mRMR-rf | 78.3 ± 0.9 | 77.6 ± 0.9 | 98.1 ± 0.2 | 83.4 ± 0.9 |
| | Block HSIC Lasso | 79.6 ± 0.2 | 78.1 ± 0.2 | 98.4 ± 0.1 | 85.2 ± 0.3 |
| | **YOTO (FS only)** | 85.3 ± 0.9 | **85.1 ± 1.3** | **98.7 ± 0.1** | 90.3 ± 0.6 |
| Integrated | PERSIST | 76.5 ± 0.7 | 85.0 ± 1.1 | 95.8 ± 0.1 | 78.1 ± 1.0 |
| | **YOTO** | **86.0 ± 0.4** | **85.1 ± 0.7** | 80.9 ± 0.2 | **90.9 ± 0.5** |

Table 9 reports the mean and standard deviation of the F1 score, accuracy, AUROC, and AUPRC on the COVID-PBMC dataset for classifying `celltypes5`. As before, our model outperforms the baselines in terms of F1 and AUPRC, which better account for class imbalance, while also achieving higher accuracy.

### A.2.1 Multi-Task Random Forest for Unsupervised Baselines

For the unsupervised gene selection baselines, employing a multi-task random forest is possible, since the same selected subset is shared across targets. Table 10 compares the F1-scores of training an MTL Random Forest on the selected subsets of Seurat and Seurat v3 for all considered subset sizes on the VISp dataset. We note that STL outperforms MTL in each case. Training one dedicated RF for each task clearly improves the results rather than having one for each.

### A.3 Implementation Details

All experiments use an 80-20 train-test split for both datasets. Results are averaged over three random seeds. All models are trained on the full training set.

**Seurat and Seurat v3.** We evaluate two variants of the Seurat algorithm using the `scanpy` Python package. Gene selection is performed using the `highly_variable_genes` method with `flavor="seurat"` and `flavor="seurat_v3"`, respectively, selecting the top $k$ genes.

**mRMR.** We evaluate two variants of the minimum Redundancy Maximum Relevance criterion using the `mrmr` package. mRMR-f uses the F-statistic to measure relevance, while mRMR-rf relies on Random Forest feature importances. In both cases, feature redundancy is quantified using Pearson correlation.

Table 10: F1-scores of MTL and STL Random Forest classifiers trained on subsets selected by Seurat and Seurat_v3 across different subset sizes on the VISp dataset.

| Task | Model | Setting | 16 | 32 | 64 | 128 | 256 |
|---|---|---|---|---|---|---|---|
| cell_types_25 | Seurat | MTL | 0.272 | 0.352 | 0.502 | 0.599 | 0.644 |
| | | STL | 0.340 | 0.483 | 0.662 | 0.774 | 0.795 |
| | Seurat v3 | MTL | 0.157 | 0.245 | 0.364 | 0.456 | 0.497 |
| | | STL | 0.194 | 0.319 | 0.492 | 0.655 | 0.749 |
| cell_types_50 | Seurat | MTL | 0.169 | 0.226 | 0.338 | 0.423 | 0.465 |
| | | STL | 0.202 | 0.323 | 0.503 | 0.654 | 0.712 |
| | Seurat v3 | MTL | 0.090 | 0.140 | 0.257 | 0.322 | 0.388 |
| | | STL | 0.112 | 0.203 | 0.365 | 0.506 | 0.619 |
| cell_types_98 | Seurat | MTL | 0.090 | 0.135 | 0.232 | 0.318 | 0.357 |
| | | STL | 0.123 | 0.220 | 0.395 | 0.552 | 0.637 |
| | Seurat v3 | MTL | 0.047 | 0.084 | 0.146 | 0.219 | 0.250 |
| | | STL | 0.059 | 0.119 | 0.255 | 0.390 | 0.506 |

**Block HSIC Lasso.** We use the Block HSIC Lasso implementation from `pyHSICLasso` with block parameter $B = 5$ and permutation parameter $M = 1$.

**Downstream Classifier** For all two-stage baselines, we train a Random Forest classifier on the selected gene subsets. To account for the class imbalance in both datasets, we apply SMOTE oversampling (Chawla et al., 2002) to the training data when training the classifier. The classifier uses 500 trees and a minimum leaf size of 15.

**PERSIST.** PERSIST is trained end-to-end following the original implementation; we report the internal classifier performance at the final epoch (n_epochs=1000). The hyperparameters used for PERSIST are the ones provided in the original code and reported in Table 11.

Table 11: PERSIST's hyperparameters.

| Batch size | Hid dim | Hid layers | Learning rate | Optimizer | Temp (init→final) |
|---|---|---|---|---|---|
| 64 | 128 | 2 | $1 \times 10^{-3}$ | Adam | 10.0→ 0.01 |

**YOTO.** YOTO is trained end-to-end using the proposed differentiable feature selection mechanism in a multi-task setting (except for Table 4 where it is trained only on the task cell_type_25). All results are reported at the end of training (n_epochs=1000) for close comparison to PERSIST, averaged over three seeds (seed∈ $\{0, 1, 2\}$). We use the AdamW optimizer for all experiments. The remaining hyperparameters used for YOTO are reported in Table 12. For each experiment, most hyperparameters are fixed, However, some training hyperparameters (learning rate and temperature schedule) depend on the target subset size $k$ and are therefore selected via a simple parameter sweep. The corresponding configurations are reported in Tables 13 and 14. Temperature values are reported as initial and final values for the annealing schedule.

| Experiment | Batch size | Hid dim | Hid layers | Warm-up epochs | Anneal epochs | Temp (init→final) |
|---|---|---|---|---|---|---|
| Figure 2 / Table 6 | 128 | 128 | 3 | 20 | 100 | $\{1, 10\} \rightarrow \{0.5, 0.01\}$ |
| Table 3 | 64 | 256 | 3 | 75 | 250 | $10 \rightarrow 0.01$ |
| Table 4 | 128 | 128 | 2 | 10 | 50 | $10 \rightarrow 0.01$ |

Table 12: YOTO's fixed hyperparameters for the respective experiments on the VISp dataset.

| num_k | Learning rate | Init. temp. | Final temp. |
|---|---|---|---|
| 16 | 0.005 | 1 | 0.5 |
| 32 | 0.002 | 10 | 0.01 |
| 64 | 0.004 | 10 | 0.01 |
| 128 | 0.002 | 10 | 0.01 |
| 256 | 0.001 | 10 | 0.01 |

Table 13: Hyperparameter sweep for Figure 2 / Table 6.

| num_k | Learning rate |
|---|---|
| 16 | 0.005 |
| 32 | 0.005 |
| 64 | 0.001 |
| 128 | 0.005 |
| 256 | 0.001 |

Table 14: Hyperparameter sweep for Table 4.

| Experiment | Batch size | Hid dim | Hid layers | Warm-up epochs | Anneal epochs | Temp (init→final) |
|---|---|---|---|---|---|---|
| Table 2 | 128 | 128 | 3 | 20 | 100 | $1 \rightarrow 0.5$ |

Table 15: YOTO's fixed hyperparameters for the respective experiments on the COVID-PBMC dataset. We use the same hyperparameters for all experiments on the COVID-PBMC dataset.

To stabilize the model training of YOTO, $k$ is annealed according to an exponential decay schedule. Starting with a larger $k$ prevents premature sparsification and yields a smoother optimization landscape for the ranking module, while the gradual reduction encourages progressive focus on the most informative features.

Let $k_0$ denote the total number of available features in the dataset, $k_T$ the target subset size, and $T$ the number of epochs over which the annealing schedule is applied. The decay rate is defined as

$$r = \frac{\log(k_T) - \log(k_0)}{T},$$  (8)

and the value of $k$ at epoch $t$ is computed as

$$k(t) = \max \left( k_0 \cdot e^{rt}, \; k_T \right), \qquad t = 0, 1, \dots, T.$$  (9)

The resulting value $k(t)$ is cast to an integer.

