# OpenReview forum: "You Only Train Once: Differentiable Subset Selection for Omics Data"
_TMLR — Accepted by TMLR_

### Review · Reviewer_DxXq · 2025-12-30

**Summary Of Contributions:**

The paper proposed a genomic multi-task learning framework that incorporates both feature selecetion and model training into a fully differentiable pipeline. Knowledge transfer between different datasets and tasks in this approach is demonstrated.
By leveraging a differentiable ranking mechanism inspired by the Plackett-Luce model and the Gumbel-Softmax trick, the framework YOTO establishes a closed feedback loop where prediction losses directly optimize a sparse, discrete binary mask during the forward pass. This end-to-end approach eliminates the need for post-hoc feature attribution or retraining separate downstream classifiers, as the model enforces sparsity throughout training to ensure only the selected $k$ genes contribute to inference. Furthermore, the multi-task architecture enables YOTO to exploit shared biological signals across related objectives, such as cell type granularity and disease status, improving generalization and allowing partially labeled datasets to inform one another. Experimental results on single-cell RNA-seq and spatial transcriptomics datasets demonstrate that YOTO consistently matches or outperforms state-of-the-art baselines like PERSIST and Seurat v3 across various gene subset sizes, particularly excelling in performance metrics for imbalanced data such as F1-score and AUPRC.

**Audience:**

Yes

**Audience Explanation:**

The proposed approach has some different features compared to baselines, as shown in table 1. Besides, the fully differentiable framework makes it easier to use.

**Claims And Evidence:**

No

**Claims Explanation:**

It is unclear how partially labeled datasets can help one another:

- Can unlabeled data in partially labeled datasets be utilized? Since the framework relis on labels during training, only labels data are useful during training.

- As a framework for multi-task learning, part of the filtered genes can represent some shared knowledge. However, this benefit comes from data used in training, i.e., labeled data. It seems not related to the partially labeled dataset-related claim.

**Requested Changes:**

- Critical to securing my recommendation for acceptance:

It seems unlikely to find a subset that is best for each task, so the identified subset by this approach is suboptimal. The motivation of developing a model for multiple tasks rather than select one subset for each task is not convincing. The training for such tasks are not too expensive to make a trade-off between training multiple models and performance.

Can the authors evaludate YOTO on single-task settings? It would be important to compare that with the current multi-tasks resutls.

Other points are illustrated in the blocks above.

- Simply strengthen the work in my view:

In fig. 1, it is unclear where the scores come from.

It would be better to discuss why YOTO performs better in the setting of large number of genes.

---

> ### Author Response · Authors · 2026-01-27
>
> We would like to thank the reviewer for the constructive feedback and for taking the time to review our paper. We are pleased that they appreciate the full differentiability of our approach and its distinction from baseline methods.
>
> We address each question below:
> - **unclear how partially labeled datasets can help one another:**
> Prior work has shown that multi-task learning (MTL) with partially annotated data (not every sample is labeled for every task) can outperform single task learning that uses only the samples labeled for a given task [1, 2]. This can be explained by the shared representation learned by a shared encoder, which allows tasks to benefit from each other even when supervision is incomplete. This benefit does not arise from using unlabeled data for a task, but from supervision provided by other tasks on the same samples, which updates the shared encoder and enables information sharing. We acknowledge that this claim is based on general findings in the MTL literature rather than being explicitly demonstrated in our experiments, thus in the updated version of the manuscript, we have moved this discussion to the conclusion section as a direction for future work.
>
> - **MTL and partially labeled datasets:**
> An important assumption (which we have clarified in the new version of the manuscript) is that tasks share underlying biological structure, making a shared subset of genes meaningful. We agree with the reviewer that a single subset is unlikely to be optimal for unrelated tasks. However, many biologically relevant tasks (e.g., related cell-type classifications) exhibit substantial overlap. Overall, our goal is to demonstrate the versatility of an end-to-end, differentiable framework that can operate in both single-task and multi-task settings and identify shared informative features when such a structure exists. We deemphasized the claims about reducing training costs.
>
> - **Evaluation of YOTO in the single-task setting:**
> Results of YOTO in the single task setting are provided in Table 9 in the Appendix in the new version of the manuscript. This shows results for the cell_type_25 task of the VISp dataset, averaged over 3 different seeds. Across the first three columns, the same architecture is used (2 hidden layers). Columns 1 and 2 differ in the selection module; notably, PERSIST employs a soft, non-binary mask, allowing non-selected genes to continue contributing to the prediction, which makes this comparison favorable to PERSIST. Columns 2 and 3 differ only in the training paradigm (STL vs MTL). The final column reports the best performance achieved for each gene subset size using a larger-capacity architecture (3 hidden layers, same embedding dimensions), as additional model capacity may be required to effectively optimize the more complex multi-task objective.
> Overall, MTL achieves performance comparable to, or slightly below, STL when using the same architecture, consistent with prior work [3] and likely due to increased optimization difficulty. While MTL is an added flexibility of our framework and can be beneficial in certain settings, it is neither strictly necessary nor always optimal for this task. We have added section A.2.1 in the Appendix to clarify this in the revised manuscript.
>
> - **Figure clarification:** We thank the reviewer for the suggestion and we have added an arrow going from the loss to the subset score to highlight that the scores are derived from the loss.
>
>
> - **Why YOTO performs better when considering larger sets of genes:** We believe that YOTO tends to perform better with larger gene subsets because its discrete selection mechanism is well-suited to capture complementary and interacting gene sets rather than relying on a few individually predictive markers. As the subset size increases, the model can better exploit coordinated biological signals and the shared representation learned end-to-end compared to two-stage approaches. Moreover, larger subsets reduce the sensitivity of performance to individual gene choices, making optimization more stable and allowing the model to better leverage the shared representation learned by the encoder. In contrast, at very small subset sizes, the selection problem becomes highly constrained, and small differences in gene choice can lead to large performance fluctuations, which limits the advantage of discrete end-to-end selection. We thank the reviewer for this insight and we have added this rationale to the discussion.
>
> We believe that these clarifications about the MTL setting, partially labeled datasets and the added comparison to the STL setting address the reviewer’s concerns. We are happy to provide further details if needed.
>
> *References*
>
> [1] Liu et al., “Semi-Supervised Multitask Learning”, 2007, Neurips
>
> [2] He et al., “On partial multitask learning”, 2020, ECAI
>
> [3] Kurin et al., “In Defense of the Unitary Scalarization for Deep Multi-Task Learning”, 2022, Neurips

---

### Review · Reviewer_ffj3 · 2026-01-15

**Summary Of Contributions:**

This paper introduces You Only Train Once (YOTO), a class of models combining end-to-end differentiable gene selection and multi-task learning, targeting single-cell transcriptomics use-cases. The method is evaluated on two transcriptomics datasets with multiple classification targets, and is compared to several classical gene selection baselines and a more recent deep learning method. While the technical components of YOTO are not novel, their application to gene expression analysis is claimed to be more efficient and generalise better than single-task, classical two-stage baselines.

**Additional Comments:**

- [Sec. 3.1] Clarify which parts are original contributions and what comes from the literature (e.g. Grover et al. 2025).
- [Sec. 3.1] Why is annealing $k$ necessary, and what is the proposed schedule?
- [Sec. 4.1] Why train models to identify patients? In a real application, wouldn't you normally want to _remove_ this variation, e.g. using an adversarial loss?
- [Sec. 4.3] All tasks are multiclass; how were the binary metrics computed and aggregated (e.g. micro vs macro average, etc.)? How were scores binarised?

**Audience:**

Yes

**Audience Explanation:**

Assuming the experimental concerns can be addressed, this benchmarking of classical and deep-learning-based sparse gene expression classifiers in the multi-task setting could be interesting to the omics community—even if YOTO is not shown to outperform the baselines.

**Claims And Evidence:**

No

**Claims Explanation:**

# Implementation details
No details are provided about model architectures nor any hyperparameters, for neither the proposed YOTO model nor any of the baselines. This undermines the credibility of presented results and precludes any reproducibility attempt.

# Baselines
My main concern is the uncomparable baselines:
1. The classical methods are evaluated with a random forest classifier, vs neural nets for PERSIST and YOTO (which may also be different, but no architecture details were given). This can be a strong confounder when comparing downstream performance.
2. Only YOTO is evaluated in a multi-task setting, even though nothing prevents multi-task training for all baseline classifiers. Indeed, even PERSIST can trivially be extended to multi-task.
3. Single-task YOTO is evaluated only in Sec. 4.4.4 using the PERSIST classifier setup, but is not compared to multi-task YOTO.

Because the central claim of the paper is that end-to-end multi-task learning enables selection of genes that are relevant for all tasks, the experiments should have evaluated the isolated effects of the selection mechanism and single- vs multi-task training.

# Selection evaluation
Given the claim that YOTO can learn a gene panel that is relevant for all tasks, I expected at least some of the following analyses, which were not explored:
- How much do the learned single-task panels overlap with each other and with multi-task?
- Does YOTO select the same genes as the established baselines?
- Does YOTO select the same genes as itself across different runs?
- Do all methods (or at least YOTO) select nested sets of genes when the panel size is varied?

# Efficiency & generalisation
The authors repeatedly refer to the computational advantage of YOTO compared to single-task baselines, but efficiency is never quantified or discussed in any detail. For realistic dataset scales, is the time spent fitting each model a practical bottleneck? The number of trained models by itself doesn't mean much if they can be trained in parallel on CPU in a few minutes (e.g. random forests).

Generalisation across tasks is also mentioned multiple times, but an apples-to-apples comparison of single- vs multi-task YOTO was not performed.

**Requested Changes:**

1. **Implementation details:** Architecture details and hyperparameters must be provided for all methods, even if it's in an appendix.
2. **Baselines:** To draw valid conclusions, the authors should compare the following methods, *all using the same neural net encoder + task head architecture* and *all trained in both single- and multi-task regimes*:
    - Dense model, i.e. YOTO without selection layer
    - Two-stage classical feature selection methods
    - End-to-end PERSIST
    - End-to-end YOTO
    - Two-stage YOTO: regular YOTO training → classifier trained from scratch using only the learned gene panel. This would validate purely the downstream quality of the gene selection, in a way that's directly comparable to the classical baselines.
3. **Selection evaluation:** Consider analysing the overlap of selected gene panels to strengthen the impact of the story (see suggestions above). This could be implemented with agreement metrics like Cohen's/Fleiss' kappa or similar.
4. **Efficiency & generalisation:** Please try to provide estimates of computational cost, even if very approximate. Alternatively, the efficiency claims should be de-emphasised. Similarly, unless direct evidence of generalisation can be provided, such claims should be toned down.

---

> ### Author Response · Authors · 2026-01-27
> **Part 1/3**
>
> We thank the reviewer for the thorough review and are pleased that they recognize the value of our work for the omics community.
>
> We begin by clarifying two points that are relevant for the remainder of this rebuttal.
>
> - **Original contribution**:
> While Grover et al. (2019) introduce the differentiable permutation matrix that we build upon, their work does not provide a mechanism for differentiable subset selection. Our method introduces a differentiable top-k selection process combined with a binary masking mechanism to enable discrete, differentiable subset selection in the forward pass while preserving differentiability in the backward pass. This enables true end-to-end learning of sparse, discrete subsets jointly with the prediction objective. In addition, we leverage their differential sorting for choosing the top-k samples as mentioned in section 3.1. However, other works [4, 5], would be feasible too for this step. Therefore, we see our work more as a special case of the DRPM [6] and the novelty is also in the combination of the different parts and the application, instead of a pure methodological novelty in the field of differentiable relaxations.
>
> - **Metrics**:
> While training is performed in a multi-task setting by optimizing the sum of task-specific losses, evaluation is conducted separately for each task. For each task, we report the performance of its corresponding task-specific prediction head, which is a standard single-task classification metric. As a result, no aggregation across tasks is required. We have clarified this distinction explicitly in the revised manuscript.
>
> With this in mind, we address the remaining concerns.
>
> - **Implementation details**:
> We agree that implementation details are critical for transparency and reproducibility.  We have added detailed hyperparameters to the Appendix A.3 to enable reproduction of all reported results, and the public repository linked in the manuscript contains the complete codes used in our experiments.

---

> > ### Author Response · Authors · 2026-01-27
> > **Part 2/3**
> >
> > - **Baselines**:
> > - **Isolating the classifier**:
> > We appreciate the reviewer’s suggestion to isolate the effect of the classifier, and we agree that disentangling selection quality from predictive capacity is important for a fair comparison. In our setting, however, we believe that applying the neural network architectures used in PERSIST or YOTO on top of classically selected gene sets would not achieve this goal, as in these methods the network itself plays an active role in feature selection and would introduce additional capacity for the classical methods, thereby confounding the comparison.
> > In contrast, applying a random classifier on top of the selected genes provides a fair and transparent comparison that directly evaluates the quality of the selected gene subsets themselves. We therefore included new results in Tables 2, 3, 5, 6, 7, and 8, using a random forest trained on the genes selected by YOTO, which provides a transparent, decoupled assessment of feature quality. However, we would like to highlight again that, a key advantage of YOTO is the joint optimization of selection and prediction in an end-to-end manner; decoupling the two is useful for comparison but not the intended use.
> > Notably, two-stage YOTO matches or even exceed the performance of YOTO’s integrated predictor and outperform all baselines in most settings. This demonstrates that YOTO is not only effective as an integrated model but also a strong standalone feature selection method, identifying informative gene subsets that generalize well across tasks and support strong task-specific predictive performance.
> >
> > - **Isolating the selection mechanism:**
> > Although YOTO usually uses a slightly bigger model than PERSIST (to make multi-task training more stable), in Section 4.4.4 we conduct a fair experiment where both methods use the exact same architecture and both are trained in a single-task setting. In this controlled direct comparison, the only difference between the methods is the selection mechanism. Under these conditions, YOTO still outperforms PERSIST, except when the subset size is extremely small (k = 16).
> > We attribute the weaker performance at very small panel sizes to the fully binary mask used by YOTO. In contrast, PERSIST does not apply a binary masking scheme in its selection process. While they control the sharpness of the mask via a temperature parameter, there remains a contribution of all genes (and not only the selected ones) at the end of training, which biases the evaluation. However, we could not figure out a straightforward way to introduce a binary mask for PERSIST. This makes the comparison conservative for YOTO, which still outperforms PERSIST in most settings.
> > More generally, we believe that one of the strengths of fully differentiable gene selection approaches lies in their tight coupling of selection and prediction, where the optimization of different tasks simultaneously optimizes the selection process. While tree-based methods still tend to outperform neural network based approaches for tabular data, there is increasing evidence for a change with bigger datasets, see [7]. We believe that with more and bigger datasets becoming available in biological domains, creating end-to-end optimizable pipelines is an important step toward being able to leverage bigger datasets and unlock new insights.
> >
> > - **Isolating STL vs MTL**:
> > We agree that the original STL vs. MTL comparison did not fully isolate the effect of multi-task learning, as the MTL setting used a larger architecture. Following the reviewer’s suggestion, we evaluated STL and MTL using the same architecture on the cell_type_25 task of the VISp dataset and added this experiment in Section A.2.1.. The results are reported in Table 9 of the updated manuscript.
> > We observe that MTL achieves performance comparable to or slightly below STL, consistent with prior work and likely due to increased optimization difficulty. While MTL is an added flexibility of our framework and can be beneficial in certain settings, it is neither strictly necessary nor always optimal for this task, and we have clarified this in the revised manuscript.
> >
> > - *“Only YOTO is evaluated in a multi-task setting”*:
> > We understand the reviewer's concern but evaluating the baselines in an MTL setting is not well-defined. For unsupervised gene selection methods, the selected subset is independent of downstream tasks, so MTL yields the same subset as STL. For supervised methods such as HSIC Lasso, feature selection is performed independently per task, and there is no principled way to learn a shared subset across tasks without ad hoc modifications. Similarly, PERSIST optimizes a single prediction objective at a time and does not support joint optimization of multiple task losses through a shared selection mechanism. In contrast, YOTO explicitly couples subset selection and task prediction across multiple tasks, making it the only method for which an MTL evaluation is well-defined and meaningful.

---

> > > ### Author Response · Authors · 2026-01-27
> > > **Part 3/3**
> > >
> > > - **Comparison with dense YOTO:**
> > > For the remaining comparison suggested by the reviewer, we would appreciate clarification of its motivation. A dense variant of YOTO without a selection layer removes feature selection entirely, whereas subset selection is a core objective of this work and central in multi-omics settings for interpretability and practical constraints. This comparison therefore addresses a different problem setting and we believe it does not directly evaluate the proposed contribution.
> > >
> > > - **Selection evaluation:**
> > > Based on the reviewer’s feedback and that of all the reviewers, we have added a section to evaluate the genes selected by YOTO. All results can be found in the updated manuscript. In a nutshell, we address the following questions
> > >     - *overlap across runs:* we added a new Section A.2.2 analyzing the robustness of YOTO’s gene selection across runs. Overall, approximately two thirds of the selected genes overlap in pairwise comparisons between runs, with about 40% of genes consistently selected across all seeds. This behavior is expected, as some genes are functionally interchangeable. Importantly, the consistently selected genes are significantly enriched for known biomarkers (Fisher’s exact test, odds ratio \approx 12, p < 10^-3), and a GO enrichment analysis of the top 30 biological processes shows a high level of consistency across runs, with a high GO terms Spearman correlation (0.78-0.82) between any two seeds. We also discuss the biological meaningness of the selected genes.
> > >     - *nesting behavior:* we added Section A.2.3 in which we evaluate the nestedness of the genes selected by YOTO across runs. Using the definition of recall from the information retrieval theory, we show that with increasing gene subset size this metric increases and variability decreases.
> > >
> > > - **Efficiency and generalisation:**
> > > We thank the reviewer for their advice, we have toned down these claims in the updated version of the manuscript.
> > >
> > > - **Additional comments:**
> > >
> > >     - *[Sec. 3.1]* Addressed above
> > >
> > >     - *[Sec. 3.1] Why is annealing necessary, and what is the proposed schedule?*: Annealing is a standard procedure in relaxation-based algorithms [8] and is used to progressively sharpen the differentiable ranking and encourage discrete gene selection during training. Specifically, the temperature parameter \tau in the differentiable permutation matrix controls the smoothness of the ranking: at higher temperatures, the ranking is softer and facilitates stable gradient-based optimization, while decreasing \tau makes the selection increasingly deterministic. As \tau goes to zero, the softmax converges toward an argmax, yielding a near-discrete ranking. We follow the exponential annealing schedule proposed in [8], which provides a smooth transition from exploration to discrete selection. This is described in Section 3.1 of the manuscript.
> > >
> > >     - *[Sec. 4.1] Why train models to identify patients? In a real application, wouldn't you normally want to remove this variation, e.g. using an adversarial loss?* In our setting, patient prediction is optimized jointly with cell-type prediction through separate task-specific heads and a shared encoder. As a result, the auxiliary task encourages the model to account for patient-specific variation in the shared representation, while the cell-type loss continues to drive the selection of genes that are predictive of cell identity. Including patient prediction as an auxiliary task can therefore be valuable, as it explicitly models inter-patient variability and batch-like effects, which are known confounders. In this sense, patient prediction acts as a form of regularization rather than a target of direct biological interest. That said, we acknowledge that this auxiliary objective can also encourage the retention of patient-specific genes, which may be undesirable if the sole goal is marker discovery rather than optimizing predictive performance for cell-type classification. We will clarify this distinction and its implications more explicitly in the revised manuscript.
> > >
> > >     - *[Sec. 4.3]* Addressed above
> > >
> > > We believe this addressed the reviewer’s concerns and would be happy to provide further clarification if necessary.
> > >
> > > *References*
> > >
> > > [4] Petersen, Felix, et al. "Differentiable sorting networks for scalable sorting and ranking supervision." International conference on machine learning. PMLR, 2021.
> > >
> > > [5] Prillo, Sebastian, and Julian Eisenschlos. "Softsort: A continuous relaxation for the argsort operator." International Conference on Machine Learning. PMLR, 2020.
> > >
> > > [6] Jang, Eric, Shixiang Gu, and Ben Poole. "Categorical Reparametrization with Gumble-Softmax." ICLR 2017.
> > >
> > > [7] Hollmann, Noah, et al. "Tabpfn: A transformer that solves small tabular classification problems in a second." arXiv preprint arXiv:2207.01848 (2022).
> > >
> > > [8] Sutter, Thomas, et al. "Differentiable random partition models." Advances in Neural Information Processing Systems 36 (2023).

---

> > > > ### Comment · Reviewer_ffj3 · 2026-02-09
> > > >
> > > > Thank you for the detailed response and for addressing many of my and the other reviewers' comments. Here are a few follow-up clarifications:
> > > >
> > > > - **Original contribution:** Thank you for clarifying the methodological contribution. I suggest that the authors restate this in the main text.
> > > > - **Metrics:** I meant clarifying how the metrics for each task are aggregated *across classes*, as F1 score, AUROC, and AUPRC are all binary classification metrics. Knowing whether the reported results are micro- or macro-averages is especially important under severe class imbalance (Fig. 4).
> > > > - **Baselines:**
> > > >     - **Isolating the classifier:** The added YOTO selection + RF classifier now allows fairly comparing the selected gene sets against the classical baselines. However, I still don't understand why a neural net classifier couldn't be trained on the gene sets from the classical baselines. This would clearly corroborate the benefit of end-to-end differentiable gene selection, using the same classifier capacity as YOTO and PERSIST.
> > > >     - **MTL baselines:** This question was meant to evaluate the effect of MTL vs STL on classification performance, even for fixed gene sets. Also, maybe the PERSIST codebase doesn't support MTL out-of-the box, but in theory it looks possible just like for YOTO. And scikit-learn offers MTL random forests as well.
> > > >   - **Dense model:** The suggestion to train a model on *all* available genes was simply as a reference point for performance achievable without any selection, while using the same classifier architecture. If it's better, it would show how much performance is lost by using only a subset of genes (which is justifiable in practice, as you suggest). If it's worse, it would partly support the claim of robustness of YOTO.
> > > > - **Additional comments:** [Sec 3.1] My annealing question was specifically about the panel size $k$, not the temperature $\tau$. (Though I acknowledge that the "$k$" was easy to miss in my original comment.)

---

> ### Author Response · Authors · 2026-02-13
> **Response to Reviewer ffj3**
>
> We thank the reviewer for the constructive feedback and the opportunity to further clarify our methodology and experimental design.
>
> - **Original Contribution:** We have updated the manuscript to clearly delineate our contribution.
> - **Metrics:** Thank you for clarifying our misunderstanding. We confirm that across all experiments, we report **macro averages** for F1-score, AUROC, and AUPRC. This choice was made specifically to ensure that the performance on rare cell types is not masked by more frequent populations, providing a robust assessment under the class imbalance present in transcriptomic data (Fig. 4 in the Appendix). We have added this to the manuscript.
>
> **Baselines**
> - **Isolating the classifier:** We agree that a fully symmetric “Baselines+NN” comparison would further control for model capacity. However, such an experiment would mainly assess downstream inference performance, since the feature selection step for those baselines remains decoupled from the classifier. We believe the YOTO+RF experiment (requested in the initial review) effectively addresses the concern regarding classifier capacity. By evaluating YOTO-selected gene panels with a Random Forest, we explicitly decouple our differentiable selection from the neural network’s inference capacity. Random Forests are well-established competitive baselines for tabular biological data [9,10]. Importantly, performance remains comparable when replacing the neural predictor with an RF on the same selected panels. If the observed gains were primarily driven by neural-network capacity, we would expect a substantial drop in performance when switching to RF. Instead, the stability of the results suggests that YOTO identifies intrinsically more informative gene subsets, supporting the benefit of the end-to-end selection strategy over the decoupled baselines.
>
> - **MTL baselines:** Regarding the construction of MTL baselines: The main limitation in constructing MTL baselines lies not in the choice of STL or MTL downstream classifiers, but in the gene selection algorithms themselves. As the reviewer correctly notes, multi-task random forests is supported in scikit-learn. However, their use implicitly assumes that the gene panels have been selected jointly with respect to multiple outputs. This assumption does not hold for the established supervised baselines and their publicly available implementations, which produce target-specific gene panels. As a result, each selected panel naturally requires a separate downstream classifier.
>
> For the unsupervised gene selection baselines, employing an MTL random forest is possible, since a single gene panel is shared across targets. We have added subsection A.2.2 to the Appendix, where we show the F1-scores of training an MTL Random Forest on the selected subsets of Seurat and Seurat_v3 for all considered panel sizes on the VISp dataset. We note that STL outperforms MTL in each case. Training one dedicated RF for each task clearly improves the results rather than having one for each. We will include these results in the appendix for completeness.
>
> - **Dense model:** We acknowledge that a "Dense" model (using all genes) provides a point of reference. To get a better view of this, we are currently running an experiment with fully dense models and will include the results in the revised manuscript once ready.
> In our k-annealing framework, the early stages of training essentially act as a dense model exploration. While the dense model naturally achieves higher absolute scores during that stage, the goal of YOTO is to bridge the gap between a massive input space (20,000+ genes) and a practical, low-cost diagnostic panel (16–256 genes).
> - **$k$-annealing:** Thank you for clarifying the question. We anneal $k$ according to an exponential decay schedule. We will further specify this in the implementation details, see Eq. (8) and (9).
> During early experimentation and model development, we observed that this $k$ annealing stabilized model training. By considering more genes at the start, we avoid premature sparsification, which creates an easier, more stable optimization problem for the ranking module. By starting with a large $k$ and gradually reducing it, the model can explore and train a broad set of features initially, while progressively focusing on the most informative ones as learning progresses.
>
> [9] Grinsztajn et al. "Why do tree-based models still outperform deep learning on typical tabular data?" Advances in neural information processing systems 35 (2022).
>
> [10] Shwartz-Ziv et al. "Tabular data: Deep learning is not all you need." Information fusion 81 (2022).

---

> > ### Author Response · Authors · 2026-02-16
> >
> > The F1-score results of the dense model (i.e., using all genes as input) on the VISp dataset are shown below:
> >
> >
> >
> > |                     | celltype_25 | celltype_50 | celltype_98 |
> > |----------------------------------|-----------------------|-----------------------|-----------------------|
> > | dense model         | 0.927 +- 0.010         | 0.846 +- 0.010         | 0.755 +- 0.024         |
> > | YOTO (integrated, num_k=64)   | 0.906 +- 0.009         | 0.799 +- 0.019         | 0.668 +- 0.010         |
> > |                     |                     |                     |                     |
> > | difference (mean)   | 0.021                 | 0.047                 | 0.087                 |
> >
> >
> > Overall, using the selected subset of genes results in a performance decrease of 0.021, 0.047, and 0.087 F1 points for the celltype_25, celltype_50, and celltype_98 tasks, respectively. Importantly, the performance gap increases with task complexity (i.e., with the number of cell types), suggesting that more complex classification settings may require a larger number of genes to maintain performance.
> > For example, for the celltype_25 task, YOTO achieves 0.920 +- 0.007 and 0.942 +- 0.009 when using 128 and 256 genes, respectively (see Table 5 in the manuscript). This suggests that performance can be further improved through increasing the number of selected genes and applying careful gene selection.
> >
> > We emphasize that we report here the test F1-score measured at the last checkpoint before starting the k-annealing process. The k-annealing phase begins only after the training loss has reached a plateau. We therefore consider this checkpoint to provide a fair estimate of the performance attainable when using all genes under the exact same architecture and hyperparameter configuration as the runs reported in the manuscript.
> >
> > We hope that these additional results clarify the upper bound achievable by the chosen architecture on these tasks. We are happy to provide further clarification if needed.

---

### Review · Reviewer_Hcdc · 2026-01-20

**Summary Of Contributions:**

This paper introduces YOTO (You Only Train Once), an end-to-end differentiable framework for gene subset selection in single-cell transcriptomics. The goal is to avoid the usual two-stage pipeline where feature selection is separated from the downstream predictor. YOTO learns gene scores using a differentiable ranking setup based on the Gumbel-Softmax trick and a Plackett-Luce style model, then applies a straight-through discrete top-k mask so the forward pass uses a binary subset. The model is trained with multi-task learning (MTL) to predict labels such as cell type, disease group, and patient ID, aiming to learn consensus gene signatures. On VISp and COVID-PBMC, the paper reports high F1 at k=64, for example 86.0% F1 on celltype5 and 74.9% F1 on patient ID.

Strengths
1) End-to-end unification: Feature selection and prediction are in one computational graph, so prediction loss directly updates gene scores.
2) Operational efficiency: The MTL setup supports “train once” with a single model instead of training separate models per task.
3) Hard sparsity: The straight-through top-k mask gives an explicit binary gene panel that is deployable.
4) Strong reported metrics: The reported F1 and AUPRC numbers are competitive against common baselines on the provided tasks.

Weakness:
1) Classifier Capacity Confounder: The performance gains may stem from the representational power of the Neural Network rather than the quality of the selected genes.
2) Patient ID Artifacts: Including Patient ID as an MTL target risks selecting genes that memorize donor-specific batch effects rather than generalizable biology.
3) Model selection risk: The protocol reports test performance at the final epoch after 1000 epochs for gradient-based models, without clear early stopping or checkpoint selection, this can lead to inflated results.
4) Lack of Biological Transparency: No gene lists or pathway analyses are provided to validate the claim of meaningful subsets.

**Additional Comments:**

The method is interesting and the paper is generally clear. My concerns are mostly about evaluation validity and isolating what causes the gains. In particular, please clarify whether COVID-PBMC splits are patient-disjoint, and consider adding a controlled comparison that uses the same downstream architecture for all gene subsets. If these are addressed, and the paper includes basic gene list and enrichment or marker sanity checks, the “meaningful subset” claim will be much stronger.

**Audience:**

Yes

**Audience Explanation:**

The TMLR audience, particularly those in bioinformatics and sparse DL, will find the efficiency of You Only Train Once highly relevant. The conceptual unification of MTL with differentiable sparsity to solve biological heterogeneity is a valuable methodological contribution.

**Broader Impact Concerns:**

The model predicts Patient ID with a 74.9% F1-score using only 64 genes. This demonstrates that even small gene panels are not truly anonymous, posing a re-identification risk under GDPR or HIPAA.

**Claims And Evidence:**

No

**Claims Explanation:**

The results look strong, but the evidence that YOTO is a superior selector is not convincing:
1) The main comparison mixes selection quality with predictor capacity. YOTO has a deep model, while baselines like Seurat, mRMR, HSIC Lasso are paired with a different downstream classifier. The gap could be due to non-linear modeling rather than better genes.
2) There is a leakage risk on COVID-PBMC. The paper uses an 80–20 split and also predicts patient ID and disease group. If the split is at the cell level, train and test likely share patients, which inflates patient and group performance and may bias selected genes.
3) The abstract claims meaningful subsets without showing selected gene panels, and without GO or pathway analysis, it is unclear if the model is selecting biology or confounders like stress-response or mitochondrial genes.

**Requested Changes:**

Critical to secure my recommendation
1) Freeze-the-encoder experiment: Take gene subsets from Seurat, mRMR, HSIC Lasso, then feed those subsets into the exact same neural architecture used by YOTO. This isolates selection quality from classifier capacity.
2) Patient split clarity and ablation: Specify whether COVID-PBMC is split by patient. Add a patient-disjoint evaluation. Also report results with the patient ID task removed to test if MTL helps generalization or memorizes batch effects.
3) Biological validation: Provide gene lists for major labels at k=64 (top 10–20 per task or overall), plus a basic GO enrichment or marker overlap check to justify “meaningful” subsets.
4) Reproducibility: Provide a clear repo pointer with exact scripts and configs for each table, ideally with a commit hash, and confirm Table 4 is reproducible.

Non-critical but would strengthen the work
1) Latent space visualization: UMAP or t-SNE of the encoder representations, colored by cell type and group, to support the predictive gains qualitatively.
2) Runtime accounting: Wall-clock training time and total pipeline time including hyperparameter tuning, compared to multi-stage baselines.
3) Stability: Report overlap of selected genes across seeds and across splits (for example Jaccard) to show the selector is stable.

---

> ### Author Response · Authors · 2026-01-27
>
> We thank the reviewer for the careful and detailed review, and for recognizing the methodological relevance of YOTO for the bioinformatics and sparse deep learning communities. We appreciate the constructive feedback and address the main concerns below.
>
> - **Classifier capacity confounder/Freeze the encoder experiment:**
> We agree that disentangling selection quality from predictor capacity is important. However, applying the same neural architecture used by YOTO or PERSIST on top of genes selected by classical benchmarks would not constitute a fair comparison, as in these methods the network itself participates in feature selection and would introduce additional capacity beyond the classical selector. That said, we agree with the reviewer that evaluating the selected gene subsets themselves is informative. In response, we have added an explicit two-stage evaluation in which a Random Forest classifier is trained on the genes selected by YOTO (See Tables 2, 3, 5, 6, 7 and 8 in the manuscript). This provides a decoupled assessment of selection quality that is directly comparable to classical baselines and addresses the concern that gains may stem solely from neural network capacity. Overall, these new results show that YOTO selects strong subsets of genes with the two step approach often outperforming all other methods.
>
> - **Patient split clarity and ablation:**
> In our framework, patient prediction was included with the idea of explicitly modelling inter-patient variability and batch-like effects, which are known confounders, and to act as a form of regularization rather than as a target of biological interest. However, we agree that patient prediction is not an objective that is valuable biologically and acknowledge the risk of emphasizing patient-specific signals. In highly constrained regimes (e.g., very small gene subsets), jointly optimizing multiple tasks can introduce trade-offs, where patient-informative genes may be retained at the expense of optimal cell-type markers. This effect diminishes as the panel size increases and reflects a limitation of extreme sparsity rather than a failure of the approach. We have clarified this trade-off explicitly in the revised manuscript and note adversarial formulations or task weighting as promising directions for future work.
> Regarding potential data leakage, the dataset was split at the cell level rather than by patient or disease group. We do not consider this a leakage issue, as both patient and disease group are explicit prediction targets, which means that evaluating these tasks on unseen patients or groups would be ill-defined. Moreover, including the group prediction task provides additional supervision related to disease severity, encouraging the shared encoder to capture biologically relevant variation rather than leaking test information.
>
> - **Model selection and evaluation protocol:**
> We thank the reviewer for pointing out the importance of model selection. Both gradient-based models (YOTO and PERSIST) are trained for a fixed number of epochs (1000). This choice was made to remain as comparable as possible with PERSIST. In PERSIST, the soft selection mask only becomes sufficiently sharp toward the very end of training, even though non-selected genes still retain non-zero weights even at this stage, which confers an advantage during evaluation. To ensure a fair comparison, we therefore evaluated YOTO at the same training stage. While early stopping could further improve YOTO’s performance (as our runs have shown), our goal was not to maximize absolute performance, but to compare methods under similar and controlled conditions.
>
> - **Biological transparency and validation:**
> We thank the reviewer for this suggestion and we agree that biological validation strengthens the claim of meaningful subsets. Thus, we have added new sections (A.2.2 and A.2.3 in the Appendix) to analyze the selected genes in more details,, including overlap with known marker genes. We also included stability analyses (overlap across seeds) and GO term analysis to provide additional transparency regarding the learned gene panels.
>
> - **Reproducibility:**
> We agree on the importance of reproducibility and have added the full code in an anonymous repository in the initial submission. We have now added full architectural details and hyperparameters for all methods and each Table in the appendix. We confirm that Table 4 is reproducible as reported
> .
> - **Additional suggestions:**
> We thank the reviewer for the additional suggestions regarding latent space visualization and runtime accounting. We have deemphasized the strong efficiency claims.
>
> We believe these revisions strengthen the experimental validity and interpretability of the results, and address the reviewer’s concerns. We thank the reviewer again for their constructive feedback.

---

### Author Response · Authors · 2026-01-29
**General response**

We thank all reviewers for their thorough and constructive feedback. We appreciate the recognition of the methodological relevance of YOTO for the bioinformatics and sparse deep learning communities (```Hcdc```, ```ffj3```), the operational efficiency and conceptual unification of the framework (```Hcdc```, ```DxXq```), as well as the strong reported metrics on the provided tasks (```Hcdc```). We have included a detailed point-by-point response to all the reviewers' open questions in the individual threads and have updated the manuscript accordingly. To facilitate the review of these revisions, all new sections and significant changes in the updated PDF are highlighted in blue.

One of the main open concerns is the disentanglement of feature selection quality from classifier capacity. We clarify that while the end-to-end nature of YOTO is a key contribution, validating the selected features independently is crucial. In response, we have added a comprehensive two-stage evaluation in which Random Forest classifiers are trained on genes selected by YOTO. This allows for a direct comparison with classical baselines, and results demonstrate that YOTO selects highly informative subsets as it outperforms the other baselines in most settings.

We also clarified the behavior of the model regarding Single-Task (STL) versus Multi-Task Learning (MTL). We have added new experiments isolating STL and MTL performance using identical architectures. These results confirm the specific benefits of the multi-task objective while clarifying that MTL is not universally necessary; rather, it serves as a flexible feature of our framework that can be easily deployed.

Furthermore, to address concerns regarding biological transparency, we have significantly expanded the Appendix (A.2.2) to include stability analyses, gene overlap statistics across seeds, and GO enrichment analyses. Finally, we have ensured full reproducibility by adding detailed hyperparameters and architectural specifications for all methods to the Appendix (A.3).

We thank the reviewers again for their valuable feedback, which has helped improve the experimental rigor and interpretability of the paper!

---

### Decision · Action_Editor_3pTD · 2026-03-11

**Recommendation:** Accept with minor revision

**Additional Comments:**

The authors are required to make the following minor revisions before the final publication:

In the “Limitations” or “Discussion” section, include a brief discussion about the challenges of cell-level versus patient-level splits for biomarker discovery, as suggested during the review process.

To address the reproducibility issue, the authors claimed to have submitted a repository for the code. However, I don’t see it in the submission. This must be corrected before the paper can be accepted, and the code repository should be explicitly mentioned in the paper.

**Audience:**

Yes

**Audience Explanation:**

The proposed approach might be interesting to the TMLR community, especially for those focus on sparse learning and bioinformatics.

**Claims And Evidence:**

Yes

**Claims Explanation:**

The submission presents YOTO, an end-to-end differentiable framework for gene subset selection in single-cell omics. The authors effectively addressed the primary technical concerns raised by the reviewers regarding the relationship between neural network capacity and feature selection quality. Specifically, they demonstrated that the selection quality is superior to classical baselines, independent of the deep learning architecture.